 SciPost Phys. Lect. Notes 51 (2022)

# Lecture notes on Berry phases and topology

**Barry Bradlyn[1*] and Mikel Iraola[2,3]**

**1** Department of Physics and Institute for Condensed Matter Theory,
University of Illinois at Urbana-Champaign, Urbana, IL, 61801-3080, USA
**2** Donostia International Physics Center, 20018 Donostia-San Sebastian, Spain
**3** Department of Physics, University of the Basque Country UPV/EHU,
Apartado 644, 48080 Bilbao, Spain

* bbradlyn@illinois.edu

## Abstract

In these notes, we review the role of Berry phases and topology in noninteracting electron systems. Topics including the adiabatic theorem, parallel transport, and Wannier functions are reviewed, with a focus on the connection to topological insulators.

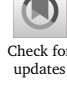

# 1 Introduction

These notes are adapted from a series of lectures given at the 2018 Topological Matter School in Donostia-San Sebastian [1]. The main focus is on Berry phases in the band theory of solids, with a particular emphasis on topological insulators and Wannier functions. We will start by first reviewing the adiabatic theorem in some generality, showing how parallel transport and holonomy in parameter space relate to the (non-abelian) Berry phase. Next, we will show how in the particular case of Bloch Hamiltonians, parallel transport of the crystal momentum can be used to calculate the electrical polarization of insulators. We will then relate this to localization and Wannier functions. We will introduce the Wilson loop to encode non-abelian holonomy along non-contractible paths in the Brillouin zone, and show how symmetries can place constraints on the Wilson loop. Using these tools, we show how the Wilson loop allows us to distinguish between topologically distinct sets of Bloch bands. Finally, we will show how topologically nontrivial bands present an obstruction to forming localized Wannier functions, and how these obstructions manifest in the Wilson loop. Exercises are referenced throughout the notes, and collected in the penultimate section.

While these notes closely follow the original lectures, we have made some modifications to accommodate this new format. Since the lectures occurred in the middle of the satellite school, they assumed some knowledge of Bloch Hamiltonians and the tight-binding method presented in prior talks. We have made some attempt to introduce these concepts here, although a reader who is completely unfamiliar with these concepts would be advised to read up on them first. A good reference for this, and for the material covered in these notes, is "Berry Phases in Electronic Structure Theory," by D. Vanderbilt [2]. Additionally, these lectures were originally followed by a discussion of how the theory of band representations, through the recently developed framework of "topological quantum chemistry," gives a unified understanding of topological crystalline phases from a real-space, Wannier function-centered point of view. A pedagogical introduction to these ideas can be found in Ref. [3–5], as well as the original literature [6–9].

# 2 Parametric Hamiltonians and Parallel Transport

We will start this section by showing that for noninteracting electrons moving in a periodic potential, the set of Bloch Hamiltonians as a function of crystal momentum $\mathbf{k}$ forms the kind of parametric family of Hamiltonian considered in the quantum adiabatic theorem. Then, we will present the formalism of adiabatic transport quite generally, where the concepts of parallel transport, Berry connection and holonomy defined in parameter space will arise. Finally, we will show how these concepts apply in a particular example: spin-1/2 in a magnetic field.

## 2.1 Parametrization of Bloch's Hamiltonian

Recall that for noninteracting electrons in a periodic potential, Bloch's theorem allows us to label each eigenstate of the Hamiltonian $H$ by a pair of quantum numbers $(n, \mathbf{k})$, where $n$ is a band-index and $\mathbf{k}$ is the crystal momentum belonging to the first Brillouin zone (BZ). The time-independent Schrödinger equation thus takes the form

$$H\psi_{n\mathbf{k}}(\mathbf{r}) = E_{n\mathbf{k}}\psi_{n\mathbf{k}}(\mathbf{r}). \tag{1}$$

Furthermore, every eigenstate $\psi_{n\boldsymbol{k}}(\boldsymbol{r})$ is a Bloch wave that can be written as

$$\psi_{n\boldsymbol{k}} = e^{i\boldsymbol{k}\cdot\boldsymbol{r}} u_{n\boldsymbol{k}}(\boldsymbol{r}),  \tag{2}$$

where $u_{n\boldsymbol{k}}(\boldsymbol{r})$ is a function with the same periodicity as the Bravais lattice of the crystal, i.e. $u_{n\boldsymbol{k}}(\boldsymbol{r}+\boldsymbol{R}) = u_{n\boldsymbol{k}}(\boldsymbol{r})$, with $\boldsymbol{R}$ a vector belonging to the Bravais lattice. By substituting Eq. (2) into Eq. (1), we can write down the Schrödinger equation for the periodic functions:

$$H(\boldsymbol{k}) u_{n\boldsymbol{k}}(\boldsymbol{r}) = E_{n\boldsymbol{k}} u_{n\boldsymbol{k}}(\boldsymbol{r}),  \tag{1a}$$

in general, where we have introduced the operator representation of the Bloch Hamiltonian,

$$H(\mathbf{k}) = e^{-i\mathbf{k}\cdot\boldsymbol{r}} H e^{i\mathbf{k}\cdot\boldsymbol{r}}.  \tag{3}$$

Often it will be convenient to work with the matrix elements of the Bloch Hamiltonian projected into some fixed basis of tight-binding orbitals. Letting $|u_{n\boldsymbol{k}}\rangle$ denote the column vector we obtain by projecting $u_{n\mathbf{k}}(\mathbf{r})$ into a fixed tight-binding basis, we can write

$$H(\boldsymbol{k})|u_{n\boldsymbol{k}}\rangle = E_{n\boldsymbol{k}}|u_{n\boldsymbol{k}}\rangle,  \tag{1b}$$

in the tight-binding approximation, where $H(\mathbf{k})$ should be understood as a matrix. In these notes, we will always use the ket notation to denote Bloch functions expanded in the space of tight-binding basis vectors to avoid confusion. Eqs. (1a),(1b) were derived by noting that states are indexed by their crystal momentum $\boldsymbol{k}$ and separating the Hamiltonian into blocks of different $\boldsymbol{k}$. However, we can equally well consider the Bloch Hamiltonian $H(\boldsymbol{k})$ as a **function** of $\boldsymbol{k}$. The set

$$\{H(\boldsymbol{k}), \ \boldsymbol{k} \in \text{Brillouin zone}\}  \tag{4}$$

then forms the sort of family of parametric Hamiltonians considered in the quantum adiabatic theorem. Although there is no notion of time at this point (which makes the discussion of adiabaticity a bit premature), we will see in Sec. 3 how adiabatic variation of $\boldsymbol{k}$ is related to response to an electric field. It will thus be beneficial for us to review some properties of adiabatic transport.

We will take a slightly more geometrical point of view than that given in introductory textbooks. This will allow us to treat the continuum and tight-binding approaches on equal footing. For more details about this approach, see Refs. [10, 11].

## 2.2 Adiabatic transport

Let us consider a family of Hamiltonians $\{H(\boldsymbol{\lambda}), \ \boldsymbol{\lambda} \in \mathcal{M}\}$ with the parameter space $\mathcal{M}$ a smooth manifold. We will take $H(\boldsymbol{\lambda})$ to have a discrete spectrum for every $\boldsymbol{\lambda}$. Furthermore, let us suppose we have a collection of $N$ states

$$\mathcal{R}(\boldsymbol{\lambda}) = \{|\psi_n(\boldsymbol{\lambda})\rangle, \ n = 1,...,N\},  \tag{5}$$

so that

$$H(\boldsymbol{\lambda})|\psi_n(\boldsymbol{\lambda})\rangle = E_n(\boldsymbol{\lambda})|\psi_n(\boldsymbol{\lambda})\rangle,  \tag{6}$$

and that there exists a $\Delta > 0$ such that for every $\boldsymbol{\lambda}$ and $|\varphi\rangle \notin \mathcal{R}(\boldsymbol{\lambda})$ satisfying $H(\boldsymbol{\lambda})|\varphi\rangle = E(\boldsymbol{\lambda})|\varphi\rangle$ we have

$$\min_n |E(\boldsymbol{\lambda}) - E_n(\boldsymbol{\lambda})| \geq \Delta,  \tag{7}$$

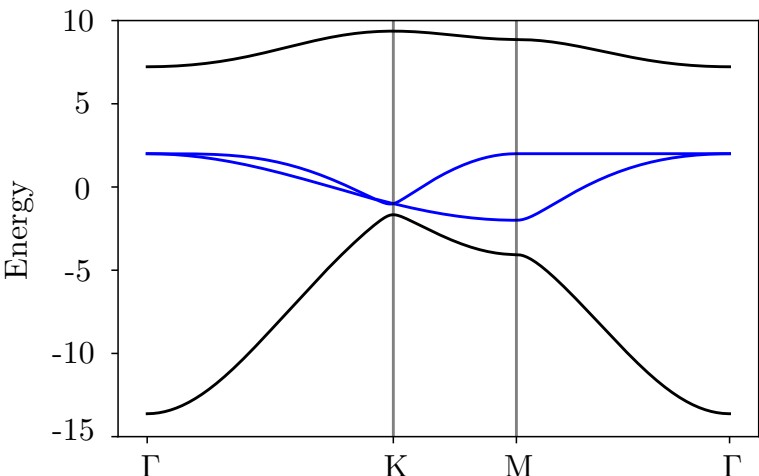

Figure 1: The concept of adiabatic transport is applicable to band structures defined in the reciprocal space of a periodic system, where the vector $k$ in the Brillouin Zone plays the role of the set of parameters $\lambda$. In this figure, we show an example of such an application: the blue bands form the family of states $\mathcal{R}(k)$ in the image of the projector $P(k)$, which are separated from states belonging to the rest of bands (black).

i.e. our family $\mathcal{R}(\lambda)$ is gapped from all other states in the spectrum for all $\lambda \in \mathcal{M}$. We can then define a family of projection operators $P(\lambda)$ with the following properties:

1. $P(\lambda)^2 = P(\lambda)$ (idempotence),

2. $[H, P(\lambda)] = 0$,

3. $P(\lambda)|\psi_n(\lambda)\rangle = |\psi_n(\lambda)\rangle$, $\forall \lambda \in \mathcal{M}$ and $|\psi_n(\lambda)\rangle \in \mathcal{R}(\lambda)$,

4. rank $P(\lambda) = N$.

It can be shown (See Exercise 1) that such an operator $P(\lambda)$ can be written as

$$P(\lambda) = \frac{1}{2\pi i} \oint_{C(\lambda)} \frac{\mathbb{I}}{z - H(\lambda)} dz, \tag{8}$$

where $C(\lambda)$ is a contour in the complex plane enclosing all the $E_n(\lambda)$ and no other eigenvalues of $H(\lambda)$. Note now that since $\mathcal{R}(\lambda)$ span the image $\text{Im}[P(\lambda)]$ of the projector $P(\lambda)$, we can equivalently define our set of states by the projector $P(\lambda)$. What is more, the fact that $P(\lambda)$ was derived from a Hamiltonian is not particularly relevant. What is important is that we have a well defined family of states. The formalism we will introduce below holds equally well for projectors onto families of quantum states, projectors onto the tangent spaces of manifolds, as well as more general fiber bundles [12, 13]. In particular, for the analysis of band structures in insulators, $P(\lambda)$ may be chosen as the projector onto the wave functions of the filled bands at each **k** point in the Brillouin Zone as indicated schematically in Fig. 1.

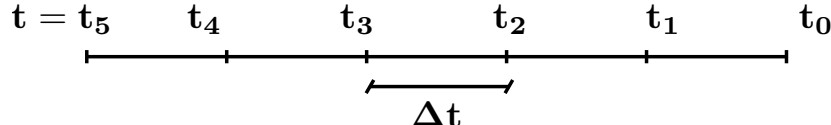

Figure 2: Time-slicing adopted in the discretization of the path ordered exponential of Eq. (14), for $N = 5$. Note that time increases to the left.

In the language of projectors, the well-known adiabatic theorem takes a particularly geometrical form, first illustrated by Kato [14]: Consider a path $\lambda(t)$, $t \in [0, \tau]$ in the parameter space $\mathcal{M}$, such that $\tau \to \infty$ ($\tau\Delta \gg 1$) for fixed endpoints of the path. Notice that $t$ can be interpreted as a scalar playing the role of time. Then the quantum adiabatic theorem is the statement that the exact projector $P(t)$ at time $t$ is approximately equal to our projector $P(\lambda(t))$ onto the space spanned by $R[\lambda(t)]$:

$$P(t) = U(t)P(0)U^\dagger(t) \approx P(\lambda(t)), \tag{9}$$

where $U(t)$ is the time-evolution operator corresponding to the Hamiltonian $H(\lambda(t))$. Following Kato, let us introduce an adiabatic evolution operator $U_A(t)$, satisfying

$$P(\lambda(t)) = U_A(t)P(0)U_A^\dagger(t). \tag{10}$$

We will see that it is possible to define $U_A(t)$ such that it is determined entirely from the geometry of $P(\lambda)$. In order to show this, let us differentiate in both sides of Eq. (10) and apply the identity $U_A\dot{U}_A^\dagger = \frac{d(U_AU_A^\dagger)}{dt} - \dot{U}_AU_A^\dagger$. This yields

$$
\begin{aligned}
i\dot{P} &= i\left[\dot{U}_AP(0)U_A^\dagger + U_AP(0)\dot{U}_A^\dagger\right] = i\left(\dot{U}_AU_A^\dagger U_AP(0)U_A^\dagger - U_AP(0)U_A^\dagger\dot{U}_AU_A^\dagger\right) \\
&= \left[i\dot{U}_AU_A^\dagger, P\right].
\end{aligned}
\tag{11}
$$

Due to the idempotence of projector $P(\lambda)$, it can be shown that $P\dot{P}P = 0$ (see Exercises 2 and 3). Using this, we can verify that Eq. (11) is satisfied if we take $\dot{U}_AU_A^\dagger = [\dot{P}, P] + f(H(\lambda))$, for any arbitrary function $f$. Choosing $f(x) = x$ results in an equation for the adiabatic evolution operator that correctly accounts for the dynamical phase that individual states acquire during evolution [11, 15]. For this choice it is possible to derive[1] an expression of the difference between $U_A(t)$ and $U(t)$:

$$U_A^\dagger(t)U(t) - \mathbb{I} = \mathcal{O}(1/\tau). \tag{12}$$

When the time needed to complete the path in parameter space is large ($\tau \to \infty$), the evolution governed by $U(t)$ may be substituted by the adiabatic evolution described by $U_A(t)$.

Since we are interested primarily in the behavior of the subspace $\mathcal{R}(\lambda)$, we can make the simpler choice $f = 0$. This leads to the following differential equation for the adiabatic evolution operator,

$$\dot{U}_A = [\dot{P}, P]U_A \equiv \mathcal{A}_s U_A. \tag{13}$$

The solution of this equation is a path-ordered exponential:

$$U_A(t) = \mathcal{P}e^{\int_0^t \mathcal{A}_s dt'} \equiv \lim_{\Delta t \to 0} e^{\mathcal{A}_s(t_N)\Delta t}e^{\mathcal{A}_s(t_{N-1})\Delta t}\ldots e^{\mathcal{A}_s(t_0)\Delta t}, \tag{14}$$

---

[1]A rigorous proof lies outside the trajectory of these lectures, but can be found in Ref. [11]

where $t_N = t$, $t_j = j\Delta t$ and $j = 0, ..., N$ (this notation for the time-slicing is shown in Fig. 2). Note that since

$$\mathcal{A}_s dt = [\partial_\lambda P, P] \cdot \dot{\boldsymbol{\lambda}}(t)dt = [\partial_\lambda P, P] \cdot d\boldsymbol{\lambda}, \tag{15}$$

the integral expression for $U_A(t)$ is **independent** of the rate at which $t$ is varied, and only depends on the particular adiabatic path from the initial point $\boldsymbol{\lambda}_i = \boldsymbol{\lambda}(t = 0)$ to the final point $\boldsymbol{\lambda}_f = \boldsymbol{\lambda}(t)$ in parameter space. Thus, $U_A$ is a **geometric** quantity and it is purely determined from the geometry of the projectors $P(\boldsymbol{\lambda})$. Nevertheless, the Hamiltonian must evolve slowly with $\tau\Delta \gg 1$, else correction terms in Eq. (12) will be non-negligible.

This discussion becomes even nicer if we restrict our attention to states $|\varphi(\boldsymbol{\lambda})\rangle \in \text{Im}[P(\boldsymbol{\lambda})]$. Consistent with the facts that $P(\boldsymbol{\lambda}) = U_A P(0) U_A^\dagger$ and $P(\boldsymbol{\lambda})|\varphi(\boldsymbol{\lambda})\rangle = |\varphi(\boldsymbol{\lambda})\rangle$, we have

$$|\varphi(\boldsymbol{\lambda})\rangle = U_A |\varphi_0\rangle. \tag{16}$$

This looks like the time-evolution of states in the Schrödinger picture, with $U_A$ playing the role of time-evolution operator. Differentiating this expression yields:

$$\partial_\lambda |\varphi(\boldsymbol{\lambda})\rangle = \partial_\lambda U_A |\varphi_0\rangle = [\partial_\lambda P, P] |\varphi(\boldsymbol{\lambda})\rangle = [\partial_\lambda P, P] P |\varphi(\boldsymbol{\lambda})\rangle = (\partial_\lambda P) |\varphi(\boldsymbol{\lambda})\rangle, \tag{17}$$

and hence

$$[\partial_\lambda - (\partial_\lambda P)P] |\varphi(\boldsymbol{\lambda})\rangle = P \partial_\lambda |\varphi(\boldsymbol{\lambda})\rangle = 0. \tag{18}$$

Eq. (17) is known as the **parallel transport** equation. It tells us that under adiabatic evolution, the projection of states into the subspace of interest does not change; thus it is a generalization of transporting a vector along a curve such that the angle of the vector with a line tangent to the curve is constant. The quantity $(\partial_\lambda P)P$ (or equivalently $[\partial_\lambda P, P]P$) is known as the **adiabatic (Berry) connection**, analogous to the Christoffel Levi-Civita connection in General Relativity. Note also that the adiabatic connection is precisely $\mathcal{A}_s$ from Eq. (15).

The operator form of the connection is closely related to the more conventional form, which expresses Eq. (17) in a fixed coordinate system: Let $|\psi_n(\boldsymbol{\lambda})\rangle$ be a basis for $\text{Im}[P(\boldsymbol{\lambda})]$, with $n = 1, ..., N$, so that $P(\boldsymbol{\lambda}) = \sum_{n=1}^N |\psi_n(\boldsymbol{\lambda})\rangle\langle\psi_n(\boldsymbol{\lambda})|$. Then, writing $|\varphi(\boldsymbol{\lambda})\rangle = \sum_{n=1}^N a_n(\boldsymbol{\lambda})|\psi_n(\boldsymbol{\lambda})\rangle$, we have from the parallel transport equation (17) that

$$\begin{aligned}
0 &= \partial_\lambda |\varphi\rangle - (\partial_\lambda P) |\varphi\rangle \\
&= \sum_{n=1}^N \left[ (\partial_\lambda a_n) |\psi_n\rangle + a_n |\partial_\lambda \psi_n\rangle - a_n |\partial_\lambda \psi_n\rangle - \sum_{m=1}^N |\psi_m\rangle \langle \partial_\lambda \psi_m |\psi_n\rangle a_n \right] \\
&= \sum_{n=1}^N \left[ \partial_\lambda a_n + \sum_{m=1}^N \langle \psi_n |\partial_\lambda \psi_m\rangle a_m \right] |\psi_n\rangle = 0,
\end{aligned} \tag{19}$$

where we have applied the relation $\langle\partial_\lambda \psi_m |\psi_n\rangle = -\langle\psi_m |\partial_\lambda \psi_n\rangle$ to go from the second line to the third line, and we have suppressed the explicit dependence on $\boldsymbol{\lambda}$ of coefficients and states for the sake of clarity. Since the states $\{|\psi_n(\boldsymbol{\lambda})\rangle\}$ form a basis of the subspace $\text{Im}[P(\boldsymbol{\lambda})]$, they are linearly independent, such that a linear combination of the $|\psi_n(\boldsymbol{\lambda})\rangle$ can be zero only if all the coefficients are zero. The parallel transport equation then implies

$$\partial_\lambda a_n - i \sum_{m=1}^N A_{nm}(\boldsymbol{\lambda}) a_m = 0, \tag{20}$$

where $A_{nm}(\lambda) = i \langle \psi_n | \partial_\lambda \psi_m \rangle$ is the usual Berry connection. Whether we use $\mathcal{A}_s$ or $A_{nm}$ depends on whether we view our adiabatic transformation as acting on basis vectors or coordinate functions: When we use $\mathcal{A}_s$, we view the adiabatic transformation as a unitary operator on the (basis) states of our Hilbert space. Contrarily, when we use $A_{nm}$, we view the adiabatic transformation as a matrix acting on the coordinate vector for a state in the space

$$\mathcal{R} = \bigcup_\lambda \text{Im}[P(\lambda)]. \tag{21}$$

Both approaches contain equivalent information when restricted to the subspaces $\text{Im}[P(\lambda)]$. However, note that we must find a differentiable basis for $\text{Im}[P(\lambda)]$ in order to define $A_{nm}(\lambda)$, while no such choice is needed to define $\mathcal{A}_s(\lambda)$.

In terms of coordinates, we can solve Eq. (20) to find:

$$a_n(\lambda) = \left[ \mathcal{P} e^{i \int_0^\lambda A(\lambda') \cdot d\lambda'} \right]_{nm} a_m(0) \equiv W_{nm}(\lambda) a_m(0). \tag{22}$$

Combining with Eq. (17), we have:

$$\langle \psi_n(\lambda) | U_A | \psi_m(0) \rangle = W_{nm}(\lambda). \tag{23}$$

The matrix $W$ is not invariant under $U(N)$ basis rotations, as we need to choose a basis to define it. In fact, given a basis transformation $\mathcal{U}(\lambda) | \psi_n(\lambda) \rangle = | \psi'_n(\lambda) \rangle$, we have:

$$W'_{nm}(\lambda) = [\mathcal{U}^\dagger(\lambda) U_A \mathcal{U}(0)]_{nm}. \tag{24}$$

Nevertheless, if we consider a **closed path** with $| \psi_n(\lambda) \rangle = | \psi_n(0) \rangle$, then we see that the transformation law for the matrix $W$ reduces to a similarity transformation. This implies that for closed paths the spectrum of $W$ is basis independent. We call the matrix $W$ for a closed path the holonomy of the adiabatic connection around that path. We will revisit this when we look at polarization in Sec. 3.

For convenience, we define the operator $P(\lambda) U_A P(0) \equiv \mathcal{W}(\lambda)$, which implements the parallel transport on $\mathcal{R}$. Eq. (23) shows that $\mathcal{W}(\lambda)$ and $W(\lambda)$ share the same nonzero spectrum in the fixed basis[2] $\{ | \psi_n(\lambda) \rangle \}$. Furthermore, $W_{nm}$ can be understood as a matrix element of $\mathcal{W}$ in the subspace $\mathcal{R}$.

To conclude our general discussion, we will define a particularly useful representation of $\mathcal{W}(\lambda)$. First, note that since $P(\lambda) = U_A(\lambda) P(0) U_A^\dagger(\lambda)$, we can write

$$\mathcal{W}(\lambda) = P(\lambda) U_A(\lambda) P(0) = U_A(\lambda) P(0) U_A^\dagger(\lambda) U_A(\lambda) P(0) = U_A(\lambda) P(0). \tag{25}$$

By taking a derivative and using Eq. (13) for $U_A$, we deduce that $\mathcal{W}(\lambda)$ satisfies the differential equation

$$\partial_\lambda \mathcal{W}(\lambda) = [\partial_\lambda P, P] \mathcal{W}(\lambda), \text{ with } \mathcal{W}(0) = P(0). \tag{26}$$

Looking at this differential equation and comparing it to $\partial_\lambda U_A(\lambda) = [\partial_\lambda P, P] U_A(\lambda)$, one might think that $\mathcal{W}$ and $U_A$ should be the same operator. However, since the initial conditions for $\mathcal{W}$ and $U_A$ are different, this is not the case. To find an expression for $\mathcal{W}$, let us first note that the infinite product

$$V(\lambda) = \lim_{\Delta\lambda \to 0} P(\lambda) P(\lambda - \Delta\lambda) P(\lambda - 2\Delta\lambda) \dots P(\Delta\lambda) P(0) \equiv \prod_{\lambda'}^{\lambda \leftarrow 0} P(\lambda') \tag{27}$$

---

[2]$W$ is an operator defined in the subspace of interest $\mathcal{R}$. In other words, we can write its matrix elements only for states $| \psi_n \rangle \in \mathcal{R}$. At the same time, $\mathcal{W}$ is defined in the whole Hilbert space. However, matrix elements $\langle \psi_l(\lambda) | \mathcal{W} | \psi_s(0) \rangle$, where $| \psi_l \rangle$ or $| \psi_s \rangle$ do not belong to $\mathcal{R}$, are zero.

is a solution to the ordinary differential equation (26). Indeed, this infinite product satisfies the same initial condition as $\mathcal{W}(\boldsymbol{\lambda})$, i.e. $V(\mathbf{0}) = P(\mathbf{0})$. To prove that $V(\boldsymbol{\lambda})$ also satisfies Eq. (26), we first take the derivative of $V(\boldsymbol{\lambda})$ to find

$$\partial_\lambda V = \lim_{\Delta \to 0} \frac{V(\boldsymbol{\lambda} + \boldsymbol{\Delta}) - V(\boldsymbol{\lambda})}{\boldsymbol{\Delta}} = \lim_{\Delta \to 0} \frac{P(\boldsymbol{\lambda} + \boldsymbol{\Delta}) - P(\boldsymbol{\lambda})}{\boldsymbol{\Delta}} V(\boldsymbol{\lambda}) = [\partial_\lambda P(\boldsymbol{\lambda})] V(\boldsymbol{\lambda}). \tag{28}$$

Using $P(\boldsymbol{\lambda})V(\boldsymbol{\lambda}) = V(\boldsymbol{\lambda})$ along with the result of exercise 2, we find that

$$\begin{aligned} \partial_\lambda V &= [\partial_\lambda P, P] V, \\ V(\mathbf{0}) &= P(\mathbf{0}). \end{aligned} \tag{29}$$

Since $\mathcal{W}(\boldsymbol{\lambda})$ and $V(\boldsymbol{\lambda})$ satisfy the same ordinary differential equation and initial condition, they are the same operator. Thus we conclude that

$$\mathcal{W}(\boldsymbol{\lambda}) = \prod_{\boldsymbol{\lambda}'}^{\boldsymbol{\lambda} \leftarrow 0} P(\boldsymbol{\lambda}'). \tag{30}$$

Finally, since the matrix $W$ is given by restricting $\mathcal{W}$ to the subspace $\mathcal{R}$ of states, we deduce that the matrix elements of Eq. (30) between states in $\mathcal{R}$ give $W$.

Summing up, in this section we have first derived the parallel transport equation for the adiabatic evolution of a system through a path in parameter space, and defined the operator form of the Berry connection $\mathcal{A}_s(\boldsymbol{\lambda})$ in this context. We have also derived and alternative expression of the Berry connection in terms of coefficients of the expansion of a state in $\mathcal{R}$ in terms of a fixed basis. Then, we have defined the operator $W(\boldsymbol{\lambda})$, whose spectrum in the subspace $\mathcal{R}$ is gauge invariant for closed paths in parameter space. Lastly, we showed how $\mathcal{W}(\boldsymbol{\lambda})$ can be written in terms of the projectors $P(\boldsymbol{\lambda})$.

Before moving on, let us examine how the concepts of adiabatic transport apply to a particularly useful example: a spin-1/2 system under the influence of a magnetic field.

### 2.2.1 Example: Spin-1/2 in a magnetic field

Let us consider a magnetic field $\boldsymbol{B}(t)$ of constant magnitude $|\boldsymbol{B}(t)| = B_0$, whose direction rotates adiabatically with time. This means that, if we draw $\boldsymbol{B}(t)$, it traces out a continuous path over the surface of a sphere of radius $B_0$. We can write:

$$\boldsymbol{B}(t) = B_0 \hat{B}(t), \tag{31}$$

in terms of a unit vector $\hat{B}(t)$.

Let us write this vector in polar coordinates, which will be useful when specifying paths in the parameter space of the problem (e.g. as indicated in Fig. 3):

$$\boldsymbol{B}(t) = B_0(\sin\theta(t)\cos\varphi(t), \sin\theta(t)\sin\varphi(t), \cos\theta(t)). \tag{32}$$

The dynamics of a spin-1/2 particle under the influence of this magnetic field can be described by a Zeeman-like Hamiltonian

$$H(t) = \mu \boldsymbol{B}(t) \cdot \boldsymbol{\sigma}, \tag{33}$$

where only the spin contributes to the energy. Here $\boldsymbol{\sigma} = (\sigma_x, \sigma_y, \sigma_z)$ is the vector of Pauli matrices. Note that the Hamiltonian of any gapped two-level system can be written in this form, modulo an

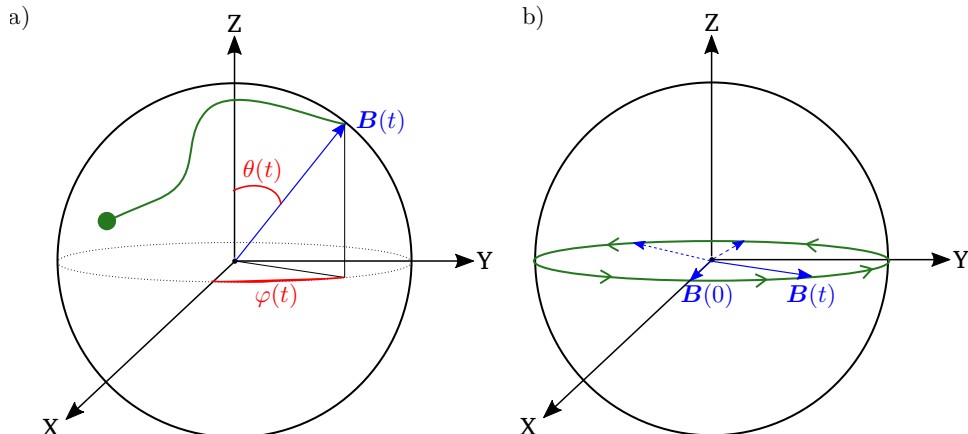

Figure 3: a) general description of the magnetic field of constant magnitude involved in the problem (blue), a general path that the field can follow over the surface of the sphere (green) and the parametrization of the path in terms of the polar angle $\theta$ and azimutal angle $\varphi$ (red). b) The particular path $(\theta(t), \varphi(t)) \in \{(\pi/2, 2\pi t) | t \in [0,1]\}$ studied in the text.

overall energy shift. Therefore, the following discussion will be applicable to any two-level system, regardless of its physical origin.

Looking at the Hamiltonian (33), we see that we can consider the direction $\hat{B}(t)$ of the magnetic field as the parameter determining a family of parametric Hamiltonians. That is, according to the notation adopted in this section, we can write $\boldsymbol{\lambda} = \hat{B}$. From Eq. (32), we see that our parameter space has dimension two: we need only specify $\theta$ and $\phi$ in order to uniquely determine $\hat{B}$. Thus, our parameter space is the two-dimensional sphere $S^2$, with coordinates

$$\mathcal{M} = S^2 = \{(\theta, \varphi) : \theta \in (0, \pi), \varphi \in (0, 2\pi)\}. \tag{34}$$

We take the subspace of interest at time $t$ to be the low-energy eigenspace of the Hamiltonian $H(t)$. The projection operator onto this subspace can be written as

$$P(t) = 1/2(\mathbb{I} - \hat{B}(t) \cdot \boldsymbol{\sigma}), \tag{35}$$

where $\mathbb{I}$ is the $2 \times 2$ identity matrix. This can be seen by considering a frame that rotates together with the field and has the $z$-axis pointing along $\hat{B}(t)$. In such a frame, the Hamiltonian takes the simple form $H(\boldsymbol{\lambda}) = -\mu B_0 \sigma_z$. Note that we can equivalently, write the projector in terms of the value $\boldsymbol{\lambda}$ reached at time t: $P(t) = 1/2(\mathbb{I} - \boldsymbol{\lambda} \cdot \boldsymbol{\sigma})$.

Now that we have the projector, we want to calculate the adiabatic evolution operator $U_A(\boldsymbol{\lambda})$ for a particular path in parameter space. We will do so in two steps. First, we will compute the Berry connection $\mathcal{A}_s$. Second, we will solve Eq. (13). Starting with the first step, we apply our definition of Berry connection to find

$$\begin{aligned}
\mathcal{A}_s^{(i)}(\boldsymbol{\lambda}) &= [\partial_{\lambda_i} P(\boldsymbol{\lambda}), P(\boldsymbol{\lambda})] \\
&= [-1/2\sigma_i, 1/2(\mathbb{I} - \sum_j \lambda_j \sigma_j)] \\
&= \frac{1}{4} \sum_j \lambda_j [\sigma_i, \sigma_j] \\
&= \frac{i}{2} \sum_{jk} \epsilon_{ijk} \lambda_j \sigma_k.
\end{aligned} \tag{36}$$

Here, $\epsilon_{ijk}$ is the Levi-Civita symbol, and the indices $i$, $j$ and $k$ in the sums run over the three Cartesian directions. In the derivation, we have made use of the commutation relation $[\sigma_i, \sigma_j] = 2i \sum_k \epsilon_{ijk} \sigma_k$ satisfied by Pauli matrices. Then, by substitution into Eq. (13), we see that the dynamics of the adiabatic evolution operator is governed by

$$\dot{U}_A(\boldsymbol{\lambda}) = \dot{\boldsymbol{\lambda}} \cdot \mathcal{A}_s(\boldsymbol{\lambda}) U_A(\boldsymbol{\lambda}) = \frac{i}{2} \sum_{ijk} \epsilon_{ijk} \dot{\lambda}_i \lambda_j \sigma_k U_A(\boldsymbol{\lambda}). \tag{37}$$

For the sake of clarity, we will write this once explictly in cartesian components:

$$\dot{U}_A(\boldsymbol{\lambda}) = \frac{i}{2} \dot{\boldsymbol{\lambda}} \cdot (\boldsymbol{\lambda} \times \boldsymbol{\sigma}) U_A(\boldsymbol{\lambda}) \tag{38}$$

$$= \frac{i}{2} [\dot{\lambda}_x (\lambda_y \sigma_z - \lambda_z \sigma_y) + \dot{\lambda}_y (\lambda_z \sigma_x - \lambda_x \sigma_z) + \dot{\lambda}_z (\lambda_x \sigma_y - \lambda_y \sigma_x)] U_A(\boldsymbol{\lambda}). \tag{39}$$

At this point, the next step is to integrate this expression to write $U_A(t)$ as a path ordered exponential. To go further, we can consider a particular path $(\theta(t), \varphi(t))$ in parameter space. Consider the following curve:

$$(\theta(t), \varphi(t)) = (\pi/2, 2\pi t), \quad t \in [0, 1], \tag{40}$$

which corresponds to starting with $\boldsymbol{B}(0)$ pointing along the positive $x$-axis, rotating its tip once around the equator, and returning to the initial point. We sketch this in Fig. 3b. Writing this path in terms of the vector $\boldsymbol{\lambda}(t)$, we have

$$\boldsymbol{\lambda}(t) = (\cos 2\pi t, \sin 2\pi t, 0). \tag{41}$$

Taking a time derivative yields

$$\dot{\boldsymbol{\lambda}}(t) = 2\pi(-\sin 2\pi t, \cos 2\pi t, 0). \tag{42}$$

Consequently, Eq. (37) becomes

$$\dot{U}_A(\boldsymbol{\lambda}) = -i\pi \sigma_z U_A. \tag{43}$$

Now, we can solve this equation, obtaining the adiabatic evolution operator:

$$U_A(t) = \exp\{-i\pi t \sigma_z\} = \cos(\pi t)\mathbb{I} - i\sin(\pi t)\sigma_z. \tag{44}$$

Let us show how $U_A(t)$ acts on the initial state $|\psi(0)\rangle = |-\rangle_x$ belonging to the subspace defined by the image of the projector $P(0)$,

$$|\psi(0)\rangle = |-\rangle_x = \frac{1}{\sqrt{2}} \begin{pmatrix} 1 \\ -1 \end{pmatrix}. \tag{45}$$

Acting with the adiabatic evolution operator, we find that the state at $t$ is

$$|\psi(t)\rangle = U_A(t)|\psi(0)\rangle = \frac{1}{\sqrt{2}} \begin{pmatrix} \cos(\pi t) - i\sin(\pi t) \\ -\cos(\pi t) - i\sin(\pi t) \end{pmatrix}, \tag{46}$$

which can be written in the basis of $\{|+\rangle_x, |-\rangle_x\}$ as

$$|\psi(t)\rangle = \cos(\pi t)|-\rangle_x - i\sin(\pi t)|+\rangle_x. \tag{47}$$

Notice that, although at $t = 1$ we reach the initial point in parameter space, the state acquires an adiabatic (Berry) phase $|\psi(1)\rangle = -|\psi(0)\rangle$. In conclusion, when we adiabatically evolve over a closed loop in parameter space, the final state may not be the same as the initial state.

Eq. (47) can be interpreted in two equivalent ways. First, we may take the perspective where we view the initial state as

$$|\psi(0)\rangle = a_+(0)|+\rangle_x + a_-(0)|-\rangle_x, \tag{48}$$

with $a_+(0) = 0$ and $a_-(0) = 1$. Comparing Eq. (47) to Eq. (48), we conclude that $U_A(t)$ has evolved the coefficients of the expansion in this way:

$$
\begin{aligned}
a_+(0) = 0 &\rightarrow a_+(t) = -i\sin(\pi t), \\
a_-(0) = 1 &\rightarrow a_-(t) = \cos(\pi t).
\end{aligned}
\tag{49}
$$

We will refer to this view of adiabatic evolution as the **active** convention: the expansion coefficients of the state evolve, but the basis stays fixed.

To understand the second point of view, let us return to Eq. (47) and attempt to express $|\psi(t)\rangle$ in terms of a basis for states in the image of $P(t)$. Looking carefully at Eq. (47), one might worry that the evolution is non-adiabatic, since $|\psi(t)\rangle$ has a component proportional to $|+\rangle_x$ which is a state outside the image of the projector of interest. To understand this, recall that $P(t)$ is the projector onto the state of lower energy of $H(t)$, i.e. onto the state $|-\rangle_{\hat{B}(t)}$. At the same time, Eq. (47) is precisely the expression of $|-\rangle_{\hat{B}(t)}$. Thus,

$$|\psi(t)\rangle = |-\rangle_{\hat{B}(t)}, \tag{50}$$

and so we see that the state $|\psi(t)\rangle$ belongs to the image of $P(t)$. In other words, when an initial state $|\psi(0)\rangle$ is evolved adiabatically, the state $|\psi(t)\rangle$ at time $t$ may have a component out of the image of a projector $P(t')$ for other times $t'$. This way of understanding the evolution is called the **passive** convention. Applying this convention is equivalent to working with a frame that rotates together with the field, keeping the positive sense of the $x$-axis pointing towards the direction of $\boldsymbol{B}(t)$.

Let us summarize both conventions explained here and mentioned previously in the text: in the active convention the coefficients of the expansion of the initial state are time-dependent, while in the passive convention the basis states taking part in the expansion are time-dependent. It is important to realize that both points of view are equivalent.

In this example, we have worked with a two-level system in which the subspace of interest is spanned by a single state. Nevertheless, the formalism of adiabatic transport is also applicable to the case in which the dimension of the image of the projectors is larger than one. In that case, there would be at least one additional eigenstate $|\psi_n(t)\rangle$ of $H(t)$ in the image of $P(t)$. The reason for including such a state may be, for example, that it shares degeneracy with the lower state $|-\rangle_{\hat{B}(t)}$ considered originally, at some point in parameter space. This occurs frequently when the states of interest are Bloch eigenstates, as we will see below.

With the general theory now established, we will move on to apply the formalism of adiabatic transport to Bloch electrons. We begin in Sec. 3 with a one-dimensional system.

## 3 Berry Phase and Polarization

In this section, we will discuss the relation between Berry phases and polarization in 1D. We will closely follow the approach of Refs. [16,17]. Let us start with the Bloch Hamiltonian Eq. (1a) for a 1D crystal with periodic potential $V(r + a) = V(r)$, where $a$ is the lattice constant:

$$H(k)u_{nk}(r) = \left[\frac{1}{2m}(p + k)^2 + V(r)\right]u_{nk}(r) = E_{nk}u_{nk}(r),\tag{51}$$

where $k$ denotes the crystal momentum defined modulo $2\pi/a$. We can consider the Brillouin zone as our parameter space $\mathcal{M}$, which is then isomorphic to the circle $S^1$. We define our projectors by means of eigenstates $|\psi_{nk}\rangle$ of the Hamiltonian:

$$P(k) = \sum_{n=1}^{N}|\psi_{nk}\rangle\langle\psi_{nk}| = \sum_{n=1}^{N}\int u_{nk}^*(r)u_{nk}(r')e^{ik(r'-r)}|r'\rangle\langle r|\,dr\,dr'.\tag{52}$$

Now, let us assume that $P(k)$ is the projector onto the $N$ "occupied" bands of an insulating crystal, and that there is a spectral gap of magnitude $\Delta > 0$ separating these bands from others in the spectrum. Consider the effect of a small uniform electric field,

$$E = -\frac{\partial}{\partial t}(-E_0 t) = -\frac{\partial A}{\partial t}.\tag{53}$$

The vector potential $A$ appears in the Hamiltonian through the minimal-coupling:

$$H(k, t) = \frac{1}{2m}(p + k - qA)^2 + V(r) \equiv \frac{1}{2m}(p + k(t))^2 + V(r) = H(k(t)),\tag{54}$$

where $k(t) = k + qE_0 t$, and $q$ is the charge of the electron (we work in units where $c = 1$). Thus, the problem of an electron moving under the influence of a constant electric field maps to a problem of evolution within a parametric family of Hamiltonians. For instance, $|qE_0|^{-1}$ plays the role of $\tau$ from the previous section; if we take $|qE_0| << \Delta$, we can apply the adiabatic theorem. Physically, this corresponds to the situation where the electric field is too weak to induce transitions to unoccupied conduction bands, so that the dynamics are restricted to the valence bands.

We find then that for an initial state $\psi_{nk}(r) = e^{ik\cdot r}u_{nk}(r)$, the final state under adabatic evolution is

$$\psi_{nk(t)}(r) = e^{ik\cdot r}W_{mn}(t)u_{mk(t)}(r),\tag{55}$$

with $W_{nm}(t) = \mathcal{P}e^{i\int_0^t A_{nm}(t')dt'}$ the matrix elements of the operator $\mathcal{W}$, and $A_{nm} = i\int_0^a u_{nk}(r)\partial_k u_{mk}(r)\,dr$. We see that the Berry phase captures the evolution of the wave functions in the presence of an electric field[3]. We can go further and relate the phase $W(t)$ to the position operator. To do so, let us consider our system to have length $L$, with periodic boundary conditions. Let us examine the average many-body position

$$\langle\mathfrak{P}\rangle = \langle\Psi_0|e^{2\pi i\hat{X}/L}|\Psi_0\rangle \equiv \langle\Psi_0|\mathfrak{P}|\Psi_0\rangle \,, \tag{56}$$

where $|\Psi_0\rangle$ is a Slater determinant ground state for an insulator in which each $|\psi_{nk}\rangle$ is occupied, and $\hat{X}$ is the many-particle position operator. In second quantization, we can write $|\psi_{nk}\rangle = c_{nk}^\dagger|0\rangle$, where

$$\begin{aligned} c_{nk}^\dagger &= \int_0^L dr\,\psi_{nk}(x)c_x^\dagger\,, \\ \{c_x, c_{x'}^\dagger\} &= \delta(x-x')\,, \\ \langle\psi_{nk}|\psi_{mk'}\rangle &= \delta_{nm}\delta_{kk'}\,. \end{aligned} \tag{57}$$

In this language, the position operator $\hat{X}$ can be written as

$$\hat{x} = \int_0^L dx\,x c_x^\dagger c_x\,. \tag{58}$$

Taking this expression into account and applying the anticommutation relations in (57), it follows that (See Exercise 5):

$$\mathfrak{P}c_x\mathfrak{P}^{-1} = e^{-2\pi ix/L}c_x\,, \tag{59}$$

and hence

$$\mathfrak{P}c_{nk}\mathfrak{P}^{-1} = \int_0^L \psi_{nk}^*(x)e^{-2\pi ix/L}c_x \equiv \tilde{c}_{nk}\,. \tag{60}$$

Using this and applying the fact that the Slater determinant ground state can be written as $|\Psi_0\rangle = \prod_{nk} c_{nk}^\dagger|0\rangle$:

$$\begin{aligned} \langle\mathfrak{P}\rangle &= \langle 0|\prod_{nk}c_{nk}\mathfrak{P}\prod_{mk'}c_{mk'}^\dagger|0\rangle = \langle 0|\prod_{nk}c_{nk}\prod_{mk'}\tilde{c}_{mk'}^\dagger|0\rangle = \det\big(\langle\psi_{nk}|\tilde{\psi}_{mk'}\rangle\big) \\ &= \det\left(\int_0^L dx\,\psi_{nk}^*(x)\tilde{\psi}_{mk'}(x)\right) = \det\left(\int_0^L dx\,u_{nk}^*e^{-ik\cdot x}u_{mk'}e^{i(k'+2\pi/L)\cdot x}\right). \end{aligned} \tag{61}$$

The determinant appears owing to the application of Wick's theorem. By considering a lattice translation $x \to x+R$, we see $\tilde{\psi}_{mk'}$ is a Bloch-wave with crystal momentum $k'+2\pi/L$. Then, conservation of crystal momentum tells us these overlaps vanish unless $k' = k-2\pi/L$, leading to

$$\langle\mathfrak{P}\rangle = \prod_k \det\left[\int_0^L dx\,u_{nk}^* u_{m(k-2\pi/L)}\right] = \det[W(2\pi/a)]\,. \tag{62}$$

---

[3]Note that, as we have considered the adiabatic evolution $U_A$ derived from Eq. (13), we have neglected the dynamical phase that can be acquired by the wave functions. See Ref. [11] for more details.

In the last equality, we have used the fact that $\det\{A\}\det\{B\} = \det\{AB\}$ to exchange the order of the product and the determinant and taken the limit $L \to \infty$. Then Eq. (30) allowed us to identify the infinite product as $W(2\pi/a)$. Therefore, we see that the gauge invariant determinant of $W$ along a closed path in the BZ is related to the mean center of charge in the unit cell. The $2\pi$ ambiguity[4] in the phase of $\det W$ descends to the polarization per unit length only being meaningful as a fraction of the electron charge $q$. This connection between the determinant of $\mathcal{W}$ and the position operator suggests that there may be a deep connection between the geometry of adiabatic evolution and localization of electrons in solids.

To explore this connection further, let us show that $\log(\langle\mathfrak{P}\rangle)$ is indeed the physical polarization density $P_e$ of the crystal, defined by Maxwell's equations to satisfy:

$$\dot{P}_e = J_{bound} = q\langle v\rangle \,. \tag{63}$$

In order to show this, we will act with $\mathfrak{P}$ on $|\Psi_0\rangle$:

$$\mathfrak{P}|\Psi_0\rangle = e^{i\gamma}\left[|\Psi_0\rangle + i\frac{2\pi}{L}\sum_{j\neq 0}|\Psi_j\rangle\langle\Psi_j|X|\Psi_0\rangle + \dots\right] =$$
$$= e^{i\gamma}\left[|\Psi_0\rangle + 2\pi\sum_{j\neq 0}|\Psi_j\rangle\frac{\langle\Psi_j|v|\Psi_0\rangle}{E_j - E_0} + \dots\right], \tag{64}$$

where we used $\langle\Psi_j|v|\Psi_0\rangle = i/L\langle\Psi_j|[H,X]|\Psi_0\rangle = i/L(E_j - E_0)\langle\Psi_j|X|\Psi_0\rangle$. Here $\gamma = \text{Im}\log\langle\Psi_0|\mathfrak{P}|\Psi_0\rangle$ is the adiabatic (Berry) phase. Note that this shows that $\mathfrak{P}|\Psi_0\rangle$ is parametrically related (via perturbation theory) to the constant electric field state treated before. Let us now assume we have a time-dependent perturbation that varies adiabatically. We can then look at the change in $\langle\mathfrak{P}\rangle$ to lowest order in perturbation theory. We have that:

$$\frac{d}{dt}\text{Im}\log\langle\mathfrak{P}\rangle = \frac{d\gamma}{dt} = \text{Im}\frac{1}{\langle\Psi_0|\mathfrak{P}|\Psi_0\rangle}\left(\langle\dot{\Psi}_0|\mathfrak{P}|\Psi_0\rangle + \langle\Psi_0|\mathfrak{P}|\dot{\Psi}_0\rangle\right). \tag{65}$$

In the adiabatic limit, $\langle\dot{\psi}_0|\psi_0\rangle = 0$ from the parallel transport equation (17), so:

$$\frac{d}{dt}\text{Im}\log\langle\mathfrak{P}\rangle = 2\pi\sum_{j\neq 0}\frac{1}{E_j - E_0}\left(\langle\Psi_j|v|\Psi_0\rangle\langle\dot{\Psi}_0|\Psi_j\rangle + \langle\Psi_j|\dot{\Psi}_0\rangle\langle\Psi_0|v|\Psi_j\rangle\right). \tag{66}$$

But the right-hand side is precisely $\langle v\rangle$ expanded to first order in perturbation theory, multiplied by $2\pi$. This shows that the Berry phase $\text{Im}\log\det W$ is, up to a multiplicative factor of $q/2\pi$, the physical polarization density. This connection between Berry phase and electronic position can be made even more precise through the exploration of Wannier functions and hybrid Wannier functions, as we will now show.

## 4 Wannier and Hybrid Wannier Functions

While Bloch's theorem tells us that the eigenstates of periodic Hamiltonians are delocalized, we know that electronic systems are built out of localized functions coming from atomic orbitals. How do

---

[4]Note that it was important that $\psi_{n(k+2\pi/a)}(r) = \psi_{nk}(r)$ in order for us to "close the loop" in the product of projectors. We will revist this shortly.

we recover these functions? Motivated by this issue, we will introduce Wannier and hybrid Wannier functions and show how the Berry phase and holonomy are connected to charge localization.

To begin, let us take a projector $P$ where $\mathrm{Im}(P)$ is spanned by the Bloch states $\{\psi_{nk}(r)\}$ for all $\mathbf{k}$ in the Brillouin zone, satisfying the boundary conditions

$$\psi_{n(k+G)}(r) = \psi_{nk}(r), \tag{67}$$

for all $\mathbf{G}$ in the reciprocal lattice. Furthermore, let us assume there exists some periodic gauge transformation $U(\mathbf{k}) \in U(N)$ such that the functions $\tilde{\psi}_{nk}(r) = U_{nm}\psi_{mk}(r)$ are analytic in $\mathbf{k}$ (and therefore differentiable in $\mathbf{k}$ to any order). Then, we can form Wannier functions via the following expressions:

$$W_{nR}(r) = \frac{1}{\sqrt{N}}\sum_{k} e^{-ik\cdot R}\tilde{\psi}_{nk}(r) \approx \frac{V}{\sqrt{N}(2\pi)^3}\int d\mathbf{k}\, e^{-ik\cdot R}\tilde{\psi}_{nk}(r), \tag{68a}$$

$$\tilde{\psi}_{nk}(r) = \frac{1}{\sqrt{N}}\sum_{R} W_{nR}(r)e^{ik\cdot R}, \tag{68b}$$

where $\mathbf{R}$ denotes vectors belonging to the Bravais lattice, $N$ the number of unit cells in the system, and $V$ is the volume. From Eq. (68b), we see that:

$$\left|\partial_{k_i}^n \tilde{\psi}_{nk}(r)\right| = \left|\frac{1}{\sqrt{N}}\sum_{R}(iR_i)^n\, W_{nR}(r)e^{ik\cdot R}\right| \leq \frac{1}{\sqrt{N}}\sum_{R}\left|R_i^n\, W_{nR}(r)\right|, \tag{69}$$

which shows that if $W_{nR}(r)$ decays faster than any power of $(r - R)$, $\tilde{\psi}_{nk}(r)$ will be smooth in $\mathbf{k}$. Hence, the smoothness of $\tilde{\psi}_{nk}(r)$ in $\mathbf{k}$ is a necessary condition for obtaining localized functions upon taking the Fourier transform. It is possible to show [18, 19] a converse to this as well: as long as $\tilde{\psi}_{nk}(r)$ is an analytic function of $\mathbf{k}$, then the Wannier functions $W_{nR}(r)$ will decay exponentially as $|r - R| \to \infty$.

Exponentially localized Wannier functions satisfy a variety of nice properties (See Exercise 7), such as

a) $\langle W_{nR}|W_{mR'}\rangle = \frac{1}{N}\sum_{kk'} e^{ik\cdot R}e^{-ik'\cdot R'}\langle\tilde{\psi}_{nk}|\tilde{\psi}_{mk'}\rangle = \frac{1}{N}\sum_{k}e^{ik\cdot(R-R')}\delta_{nm} = \delta_{nm}\delta_{RR'}$.

b) $\sum_{n=1}^{N}\sum_{k}|\tilde{\psi}_{nk}\rangle\langle\tilde{\psi}_{nk}| = \sum_{n=1}^{N}\sum_{R}|W_{nR}\rangle\langle W_{nR}|$.

c) $W_{n(R+R')}(r) = W_{nR}(r - R')$.

The first property means that Wannier functions form an orthonormal set; in the second property, we see that they span the same subspace of the Hilbert space as the band eigenstates from which they are constructed via (68a); finally, the third point means that the Wannier functions are distributed periodically through the lattice, so that it is enough to work with the Wannier functions in one unit cell (with one fixed $\mathbf{R}$). On the whole, localized Wannier functions form a complete basis that can be used to build a quantitative position space picture of the occupied subset of states in a crystal. In this spirit, Wannier functions are good candidates to study phenomena that are more intuitively understood in position space; particularly, charge localization and pumping.

As an example, let us reinterpret our expression Eq. (62) for $\langle \mathfrak{P} \rangle$ in the 1D case in terms of Wannier functions. Applying (22), we get:

$$\operatorname{Im}\log\langle\mathfrak{P}\rangle = \operatorname{Im}\log(\det W) = \oint \operatorname{Tr} A \cdot dk = i \sum_{n=1}^{N_{occ}} \int_0^{2\pi} dk \int_{cell} u_{nk}^*(r) \partial_k u_{nk}(r) \, dr \, , \qquad (70)$$

where $N_{occ}$ is the number of states in $\operatorname{Im}[P(\mathbf{k})]$. Working in the convention where

$$\tilde{u}_{nk}(r) = \sqrt{N} e^{-ik \cdot r} \tilde{\psi}_{nk}(r) = \sum_R e^{ik \cdot (R-r)} W_{nR}(r), \qquad (71)$$

(in this convention the Bloch functions $\tilde{u}_{nk}$ are normalized to one within a single unit cell) and integrating over the whole space rather than over the unit cell, Eq. (70) becomes (recall we work in units where the lattice constant is equal to one):

$$\begin{aligned}
\operatorname{Im}\log\langle\mathfrak{P}\rangle &= \frac{i}{N} \sum_{m=1}^{N_{occ}} \int dr \int_0^{2\pi} dk \sum_{RR'} e^{-ik \cdot R} W_{mR}^*(r) \, e^{ik \cdot r} \partial_k \left( e^{ik \cdot R'} e^{-ik \cdot r} W_{mR'}(r) \right) + 2\pi n \\
&= \frac{2\pi}{N} \sum_{m=1}^{N_{occ}} \sum_R \int dr \, (r-R) W_{mR}^*(r) W_{mR}(r) + 2\pi n \\
&= \frac{2\pi}{N} \sum_{m=1}^{N_{occ}} \sum_R \int dx \, x \, W_{m0}^*(x) W_{m0}(x) + 2\pi n = 2\pi \sum_{m=1}^{N_{occ}} \langle W_{m0}|r|W_{m0}\rangle + 2\pi n,
\end{aligned} \qquad (72)$$

thus, we see that $q/(2\pi)\operatorname{Im}\log\langle\mathfrak{P}\rangle$ is the polarization density in a precise sense: it gives the displacement of the average charge center from the origin of the unit cell. Here, $n$ is an integer given by the winding number [13] $2\pi i n = \oint \operatorname{Tr}\left[U^\dagger(k)\partial_k U(k)\right] dk$, of the unitary transformation that converts from the original basis $u_{nk}$ to the smooth basis $\tilde{u}_{nk}$, and shows that the Berry phase is only defined mod $2\pi$. Eq. (72) re-expresses the $2\pi$ gauge ambiguity of the Berry phase as an ambiguity of the charge center by an integer number of unit cells.

Using our knowledge of adiabatic transport, we can go further and relate the position operator to the Berry phase, without the need for the trace over occupied bands. To do so, we first introduce **hybrid Wannier functions**:

$$W_{nR_\perp}(\mathbf{r}, \mathbf{k}_\parallel) = \frac{1}{\sqrt{N_\perp}} \sum_{k_\perp} e^{-ik_\perp \cdot R_\perp} \, \tilde{\psi}_{n\mathbf{k}}(\mathbf{r}), \qquad (73)$$

which are exponentially localized in the direction denoted by $\perp$. Now, let us take a state $|f\rangle = \sum_{n\mathbf{k}} f_{n\mathbf{k}} |\psi_{n\mathbf{k}}\rangle \in \operatorname{Im}(P)$ and look at the projected position operator $PxP$. Taking matrix ele-

ments in the basis of Bloch functions, we have

$$
\begin{aligned}
\langle\psi_{n\mathbf{k}'}|Px_iP|f\rangle &= \sum_{m=1}^{N_{occ}}\sum_{\mathbf{k}}\langle\psi_{n\mathbf{k}'}|x_i|\psi_{m\mathbf{k}}\rangle f_{m\mathbf{k}} = \sum_{m=1}^{N_{occ}}\sum_{\mathbf{k}}f_{m\mathbf{k}}\int \mathrm{d}\mathbf{x}\, x_i\psi_{n\mathbf{k}'}^*(\mathbf{x})\psi_{m\mathbf{k}}(\mathbf{x}) \\
&= \frac{1}{N}\sum_{m=1}^{N_{occ}}\sum_{\mathbf{k}}\int \mathrm{d}\mathbf{x}\, f_{m\mathbf{k}}u_{n\mathbf{k}'}^*(\mathbf{x})u_{m\mathbf{k}}(\mathbf{x})e^{i(\mathbf{k}-\mathbf{k}')\cdot\mathbf{x}}\, x_i \\
&= \frac{1}{N}\sum_{m=1}^{N_{occ}}\sum_{\mathbf{k}}\int \mathrm{d}\mathbf{x}\, f_{m\mathbf{k}}u_{n\mathbf{k}'}^*(\mathbf{x})u_{m\mathbf{k}}(\mathbf{x})(-i)\partial_{k_i}\Big[e^{i(\mathbf{k}-\mathbf{k}')\cdot\mathbf{x}}\Big] \\
&= i\partial_{k_i'}f_{n\mathbf{k}'} + i\frac{1}{N}\sum_{m=1}^{N_{occ}}\sum_{\mathbf{k}}f_{m\mathbf{k}}\int \mathrm{d}\mathbf{x}\Big[u_{n\mathbf{k}'}^*(\mathbf{x})\partial_{k_i}u_{m\mathbf{k}}(\mathbf{x})\Big]e^{i(\mathbf{k}-\mathbf{k}')\cdot\mathbf{x}}.
\end{aligned}
\tag{74}
$$

Unless otherwise noted, we will use the convention that repeated indices are summed over from this point forward. Finally, we can rewrite the integral over $\mathbf{x}$ in the last term as an integral over a single unit cell, using

$$
\begin{aligned}
\int \mathrm{d}\mathbf{x}\Big[u_{n\mathbf{k}'}^*(\mathbf{x})\partial_{k_i}u_{m\mathbf{k}}(\mathbf{x})\Big]e^{i(\mathbf{k}-\mathbf{k}')\cdot\mathbf{x}} &= \sum_{\mathbf{R}}\int_{\text{cell}}\mathrm{d}\mathbf{x}\Big[u_{n\mathbf{k}'}^*(\mathbf{x}+\mathbf{R})\partial_{k_i}u_{m\mathbf{k}}(\mathbf{x}+\mathbf{R})\Big]e^{i(\mathbf{k}-\mathbf{k}')\cdot(\mathbf{x}+\mathbf{R})} \\
&= \sum_{\mathbf{R}}e^{i(\mathbf{k}-\mathbf{k}')\cdot\mathbf{R}}\int_{\text{cell}}\mathrm{d}\mathbf{x}\Big[u_{n\mathbf{k}'}^*(\mathbf{x})\partial_{k_i}u_{m\mathbf{k}}(\mathbf{x})\Big]e^{i(\mathbf{k}-\mathbf{k}')\cdot\mathbf{x}} \\
&= -i\delta_{\mathbf{k}\mathbf{k}'}A_{nm}^i(\mathbf{k}),
\end{aligned}
\tag{75}
$$

where $A_{nm}^i$ are the matrix elements of the Berry (adiabatic) connection in the $k_i$ direction between occupied bands $n$ and $m$. Putting this all together, we find

$$
\langle\psi_{n\mathbf{k}'}|Px_iP|f\rangle = i\partial_{k_i'}f_{n\mathbf{k}'} + A_{nm}^i(\mathbf{k}')f_{m\mathbf{k}'}.
\tag{76}
$$

We see that $-iP\mathbf{x}P = P\partial_{\mathbf{k}}P$ is precisely the adiabatic covariant derivative that appears in our parallel transport equation (20). Furthermore, we can also look for eigenstates of $Px_{\perp}P$, which corresponds to looking for states satisfying

$$
Px_{\perp}P|\psi\rangle = \varphi|\psi\rangle.
\tag{77}
$$

Let us take a trial solution of the form

$$
|\psi\rangle = e^{-ik_{\perp}\varphi}W_{mn}(k_{\perp})f_{n0}|\psi_{m\mathbf{k}}\rangle,
\tag{78}
$$

where $k_{\perp}$ is the component of $\mathbf{k}$ along the direction denoted by $\perp$. The matrix $W_{nm}(k_{\perp})$ is the familiar holonomy matrix, given in terms of Eq. (23), with the path given by a straight line from $\mathbf{k}_0 = 0$ to $k_{\perp}/(2\pi)\mathbf{G}_{\perp}$, with $\mathbf{G}_{\perp}$ the reciprical lattice vector in the $\perp$ direction. From the properties of $W$ we have the parallel transport equation

$$
\Big[\delta_{\ell m}\partial_{k_{\perp}} - iA_{\ell m}^{\perp}(k_{\perp})\Big]W_{mn}(k_{\perp})f_{n0} = 0.
\tag{79}
$$

The substitution of Eq. (78) in $P\partial_{k_{\perp}}P|\psi\rangle$, together with the parallel transport equation (79), yields

$$
\Big[\delta_{m\ell}\partial_{k_{\perp}} - iA_{\ell m}^{\perp}(k_{\perp})\Big]\Big[e^{-ik_{\perp}\varphi}W_{mn}(k_{\perp})f_{n0}\Big] = -i\varphi e^{-ik_{\perp}\varphi}W_{\ell n}(k_{\perp})f_{n0},
\tag{80}
$$

which matches with Eq. (77). We have shown that any function that can be expanded as (78) is a good candidate to be an eigenfunction of the projected position operator. However, we must still ensure that our choice of boundary conditions in Eq. (67) is preserved, i.e. that

$$e^{-i2\pi\varphi}W_{mn}(2\pi)f_{n0} = f_{m0}. \tag{81}$$

Thus we must choose the vector formed by the coefficients $\{f_{n0}\}$ to be an eigenvector of $W(2\pi)$ with eigenvalue $e^{2\pi i\varphi}$. In conclusion,

> The spectrum of $Px_iP$ matches the spectrum of $\dfrac{1}{2\pi}\mathrm{Im}\log W_{mn}(\mathbf{k}_0 \to \mathbf{k}_0 + \mathbf{G}_i)$.

We can go further and write down the eigenfunctions of $Px_iP$ by noting that our choice of $\mathbf{k}_0 = 0$ as the intial point for our $f_{n0}$ was arbitrary. Let us denote $W_{\mathbf{k}_0}(2\pi)$ the adiabatic evolution from $\mathbf{k} = \mathbf{k}_0$ to $\mathbf{k} = \mathbf{k}_0 + \mathbf{G}_\perp$. Let $Q(\mathbf{k}_0)$ denote the matrix containing in each column an eigenvector of $W_{\mathbf{k}_0}(2\pi)$, so that $W_{nm}^{\mathbf{k}_0}(2\pi)Q_{mj}(\mathbf{k}_0) = e^{i2\pi\varphi_j}Q_{nj}(\mathbf{k}_0)$. Furthermore, from the definition of $W_{nm}^{\mathbf{k}_0}$, we deduce that (see Exercise 8)

$$Q_{nj}(\mathbf{k}_0 + \frac{k_\perp\mathbf{G}_\perp}{2\pi}) = W_{nm}^{\mathbf{k}_0}(\frac{k_\perp\mathbf{G}_\perp}{2\pi})Q_{mj}(\mathbf{k}_0). \tag{82}$$

This implies that $e^{-ik_\perp\varphi_j}Q_{nj}(k_\perp)$ is periodic in $k_\perp$, and so satisfies Eq. (81). As a consequence, the following function is an eigenfunction of $Px_\perp P$ with eigenvalue $\varphi_j + R_\perp$:

$$W_{jR_\perp}(\mathbf{r},\mathbf{k}_\parallel) = \int \sum_{n=1}^{N_{occ}}\mathrm{d}k_\perp\, e^{-ik_\perp(\varphi_j+R_\perp)}Q_{nj}(\mathbf{k})\psi_{n\mathbf{k}}(\mathbf{r}). \tag{83}$$

Notice that the form of this function coincides with the expression of a hybrid Wannier function introduced in Eq. (73). Furthermore, since an eigenstate of $Px_\perp P$ is maximally localized in the $x_\perp$-direction, and we saw in Eq. (69) that this requires smoothness of derivatives with respect to $k_\perp$, we conclude that $Q(\mathbf{k})$ is constructed to ensure that derivatives of $\sum_n Q_{nj}(\mathbf{k})\psi_{n\mathbf{k}}(\mathbf{r})$ are smooth[5] with respect to $k_\perp$. In conclusion,

> Eigenfunctions of $Px_\perp P$ are hybrid Wannier functions localized maximally in the $x_\perp$-direction.

Consider the example of Fig. 4, where we show the holonomy and hybrid Wannier function centers for two bands in two dimensions. In (a) we show the eigenvalues of the holonomy matrix $W^{k_1,k_2=0}(\mathbf{G}_2)$–the holonomy along the $\mathbf{G}_2$ direction as a function of $k_1$. In (b) we show the location in position space of the corresponding centers of hybrid Wannier functions. These functions are maximally localized in the direction of the primitive lattice vector $\mathbf{a}_2$. When $k_1 = 0$, both centers are located a distance $\mathbf{a}_2/2$ from the center of the hexagon, corresponding to the eigenvalues $\phi/2\pi = \pm 0.5$ of the holonomy. As we start increasing $k_1$, the centers move towards the center of the hexagonal unit cell; at this point, there is a net electrical polarization in the unit cell. Finally, when $k_1/2\pi = 1/2$, both charge centers meet at the center of the hexagon.

To conclude, we have now seen how the Berry phase and holonomy encode information about charge localization via the connection to hybrid Wannier functions. First we have shown that the determinant of the adiabatic evolution operator gives the average charge center in a unit cell; then,

---

[5]The Berry connection cancels any discontinuity arising from degeneracies among states in $\mathrm{Im}(P)$.

we have gone further and we have derived the relation between the spectrum of the holonomy and the projected position operator; finally, we have concluded that the eigenfunctions of the projected position operator in a certain direction are hybrid Wannier functions maximally localized in that direction. We will conclude this section with the study of a particular 1D system, namely, the Rice-Mele chain.

## 4.1 The Rice-Mele chain

We consider a 1D inversion symmetric crystal, with lattice vector $\boldsymbol{e} = a\hat{x}$. Our basis will be formed by $s$ and $p_x$-like functions localized on each lattice site, as drawn in Fig. 5. While we will investigate the symmetry properties of $\mathcal{W}$ systematically in Sec. 5.1, here we will show that inversion symmetry has a profound effect on the Berry phase. If $U_I$ is a unitary representation of inversion, then the following properties hold:

A) $U_I P U_I^{-1} = P$,

B) $U_I P x P U_I^{-1} = -P x P$ ($x$ is odd under inversion).

Here, B) follows from the fact that the position operator $x$ is odd under inversion. This property implies that eigenvalues of $PxP$ come in pairs $\pm a\varphi/(2\pi) + \nu a$ ($\nu \in \mathbb{Z}$ appears due to the lattice ambiguity), where $\varphi$ is an eigenvalue of the holonomy matrix $W(2\pi)$. Consequently, only eigenvalues $\varphi \in \{0, \pi\}$–which correspond to hybrid Wannier functions at the center and borders of the unit cell respectively–can be unpaired. In particular, it follows for a single band that:

$$\det W(2\pi) = \pm 1 \Rightarrow \langle W_{n0}|x|W_{n0}\rangle = \begin{cases} 0 \\ a/2 \end{cases} \mod a. \tag{84}$$

In other words, inversion symmetry quantizes the polarization. Let us see this in action in our inversion symmetric chain. As we mentioned, we take as basis states $\varphi_s(r - R)$ and $\varphi_p(r - R)$ (See Fig. 5), where:

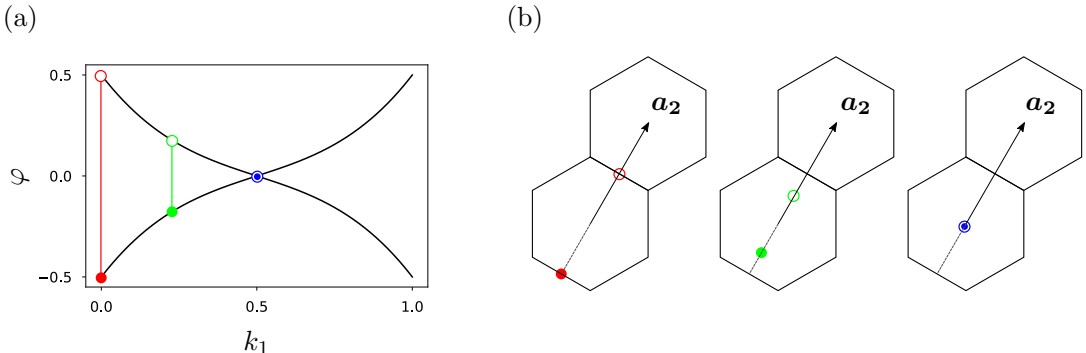

Figure 4: Eigenvalues of the holonomy $W^{k_1, k_2=0}(\mathbf{G}_2)$ for a set of 2 bands in a honeycomb lattice model [20, 21]. (a) Spectrum as a function of $k_1$, where vertical lines indicate different choices of $k_1$; two eigenvalues correspond to each choice, denoted by the solid circle and ring. (b) Interpretation of eigenvalues in terms of hybrid Wannier centers.

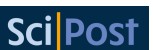
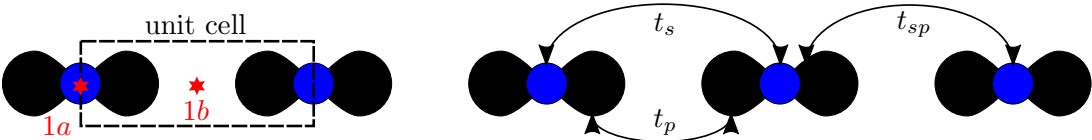

Figure 5: Schematic representation of the Rice-Mele model, where the basis formed by $s$ (blue spheres) and $p_x$-orbitals (black) is shown. The chosen unit cell is indicated with dashed lines.

$$\varphi_s(r-R) = \langle r|sR\rangle \,, \qquad\qquad (85)$$
$$\varphi_p(r-R) = \langle r|pR\rangle \,.$$

We want to construct a nearest-neighbor model respecting inversion symmetry. Let $c_{sR}$ and $c_{pR}$ be the annihilation operators for states $|sR\rangle$ and $|pR\rangle$, respectively, in second quantization. We can write the following inversion invariant terms[6]:

(a) $\sum_R \epsilon(c_{sR}^\dagger c_{sR} - c_{pR}^\dagger c_{pR})$,

(b) $\sum_R t_{sp}\big[(c_{sR}^\dagger c_{pR+1} - c_{sR}^\dagger c_{pR-1})\big] + \text{h.c.}, \quad$ where $t_{sp} = t_{sp}^{(1)} + i t_{sp}^{(2)}$,

(c) $\sum_R \sum_{\sigma=s,p} t_\sigma(c_{\sigma R}^\dagger c_{\sigma R+1} + \text{h.c.})$.

We combine these to form the tight-binding Hamiltonian

$$H = \sum_R \epsilon(c_{sR}^\dagger c_{sR} - c_{pR}^\dagger c_{pR}) + \frac{1}{2}\Big[t_{sp}\big(c_{sR}^\dagger c_{pR+1} - c_{sR}^\dagger c_{pR-1}\big) + t_{sp}^*\big(c_{pR+1}^\dagger c_{sR} - c_{pR-1}^\dagger c_{sR}\big)\Big]$$
$$+ \sum_{\sigma=s,p} t_\sigma(c_{\sigma R}^\dagger c_{\sigma R+1} + c_{\sigma R+1}^\dagger c_{\sigma R}). \qquad (86)$$

Moreover, we can take the Fourier transform $c_{\sigma k} = N^{-1/2}\sum_R e^{-ik\cdot R}c_{\sigma R}$ of the annihilation operators, which allows us to write the Hamiltonian in reciprocal space as

$$H = \sum_k \begin{pmatrix} c_{sk}^\dagger & c_{pk}^\dagger \end{pmatrix} H(k) \begin{pmatrix} c_{sk} \\ c_{pk} \end{pmatrix}, \qquad (87)$$

with

$$H(k) = \epsilon\sigma_z + t_{sp}^{(1)}\sin k\,\sigma_y + \cos k\,\sigma_z(t_s - t_p) + \cos k(t_s + t_p)\mathbb{I} + t_{sp}^{(2)}\sin k\,\sigma_x. \qquad (88)$$

For simplicity, we will take $t_s = -t_p = t/2$, as this eliminates terms proportional to the identity matrix. From the parity and distribution of orbitals in the lattice, it follows that the matrix representing inversion can be taken to be $U_I(k) = \sigma_z$, yielding

$$\sigma_z H(k)\sigma_z = H(-k). \qquad (89)$$

---

[6]This is not the most general inversion symmetric Hamiltonian that can be written with hoppings to nearest-neighbors. In particular, there is no symmetry forcing a relation between the on-site energies of each orbital. However, this simple model captures the physics we want to discuss.

In addition to inversion, we can impose time-reversal symmetry. For spinless systems (or systems in which spin-orbit coupling can be neglected), time-reversal acts in position space as complex conjugation, i.e. $\mathcal{T} = \mathcal{K}$. Then, it has the following effect on annihilation operators of Bloch states:

$$\mathcal{T}c_{\sigma,k}\mathcal{T} = N^{-1/2}\sum_R e^{ik\cdot R}c_{\sigma R} = c_{\sigma,-k}, \tag{90}$$

with the corresponding action on creation operators. This means that time-reversal symmetry imposes the condition $H(k) = H^*(-k)$ on the Hamiltonian, which requires $t_{sp}^{(2)} = 0$. At the end of the day, the spectrum of $H(k)$ with time-reversal and inversion symmetry is given by:

$$E_k = \pm\sqrt{(\epsilon + t\cos k)^2 + [t_{sp}^{(1)}]^2\sin^2 k}. \tag{91}$$

In the simple case that $t_{sp}^{(1)} \equiv t$, the model has two gapped "flat-band" limits:

(1) $t = t_{sp}^{(1)} = 0$, $\epsilon = \epsilon_0 \Rightarrow E_k = \pm|\epsilon_0|$.

(2) $\epsilon = 0$, $t = t_{sp}^{(1)} \Rightarrow E_k = \pm|t|$.

In limit (1), we have the following Hamiltonian, Bloch functions and basis states:

$$H(k) = |\epsilon|\sigma_z,$$
$$\psi_{\pm k}(r) = \frac{1}{\sqrt{N}}\sum_R e^{ik\cdot R}\varphi_{(s,p)R}(r) \equiv \varphi_{(s,p)k}(r), \tag{92}$$
$$u_{\pm k}(r) = \sqrt{N}e^{-ik\cdot r}\varphi_{(s,p)k}(r).$$

After a bit of algebra, it can be shown that the Berry connection $A_s(k)$ for the state built-up from $s$-orbitals is

$$\begin{aligned} A_s(k) &= i\int_{cell} dr\, u_{+k}^*(r)\partial_k u_{+k}(r) = N\int_{cell} dr\, e^{ik\cdot r}\varphi_{sk}^*(r)\partial_k\left[e^{-ik\cdot r}\varphi_{sk}(r)\right] \\ &= i\sqrt{N}\int_{cell}\sum_R dr\,(-i)(r-R)\varphi_{sk}^*(r)e^{ikR}\varphi_{sR}(r) \\ &= \int_{cell}\sum_{RR'} dr\,(r-R)e^{ik(R-R')}\varphi_{sR'}^*(r)\varphi_{sR}(r). \end{aligned} \tag{93}$$

Then, we can calculate the corresponding Berry phase $\gamma_s$ by integrating over the BZ:

$$\begin{aligned} \gamma_s &= \int_0^{2\pi} dk\, A_s(k) \\ &= \int_0^{2\pi} dk\int_{cell} dr\sum_{RR'}(r-R)e^{ik(R-R')}\varphi_{sR'}^*(r)\varphi_{sR}(r) \\ &= \sum_R\int_{cell} dr\,(r-R)\varphi_{sR}^*(r)\varphi_{sR}(r) \\ &= \int dr\, r\varphi_{s0}^*(r)\varphi_{s0}(r) = 0. \end{aligned} \tag{94}$$

We could have anticipated this result: since $\psi_k = \psi_{\sigma k}$, Eq. (68a) yields the Wannier function $W_{\sigma R} = \varphi_{\sigma R}$. At the same time, in 1D Wannier functions coincide with hybrid Wannier functions, thus Wannier functions are eigenfunctions of the projected position $PxP$. Note also that we could simplify our lives by working in the strict tight-binding limit in which orbitals are taken to be Dirac's deltas: $\varphi_{sR}(r) \propto \delta(r-R)$ and $\varphi_{pR}(r) \propto \delta'(r-R)$, where $\delta(r)$ and $\delta'(r)$ are even and odd under inversion, respectively; in that case, we have:

$$u_{k\sigma}(r) = e^{-ikr}\sum_R e^{ikR}\varphi_{\sigma R}(r) = \sum_R \varphi_{\sigma R}(r), \text{ independent of } k. \tag{95}$$

This means that in the strict tight-binding limit, we can evaluate the Berry connection using only the Bloch coefficients of the eigenstates (which in this case are unity). We did not have to be so drastic as to assume our basis orbitals were delta functions to get this result; more generally, we can define the **tight-binding limit** to be the case where $\langle \varphi_{\sigma R}|r|\varphi_{\sigma R'}\rangle \propto \delta_{RR'}$. In this case, the Berry phase can be evaluated entirely in terms of the Bloch coefficients. However, when the position operator has off-diagonal terms in the basis of orbitals, derivatives of Bloch functions constructed from these orbitals also contribute to the calculation of the Berry connection, so it is not enough to consider only the coefficients and their derivatives. This result is general, rather than a particular feature of the Rice-Mele chain.

The more interesting case occurs in the limit ②, where

$$H(k) = t\left(\cos k\,\sigma_z + \sin k\,\sigma_y\right),$$
$$u_{+k}(r) = \sqrt{N}e^{-ikr}e^{ik/2}\left(\cos k/2 \quad i\sin k/2\right)\begin{pmatrix}\varphi_{sk}(r)\\\varphi_{pk}(r)\end{pmatrix},$$
$$u_{-k}(r) = \sqrt{N}e^{-ikr}e^{ik/2}\left(\sin k/2 \quad -i\cos k/2\right)\begin{pmatrix}\varphi_{sk}(r)\\\varphi_{pk}(r)\end{pmatrix}. \tag{96}$$

Now, we can repeat the calculation of the Berry phase $\gamma_+$ for the state $u_{+k}$. We begin by computing the Berry connection for the column vector $|u_{+k}\rangle$ of expansion coefficients of our occupied state in the tight-binding limit $A_+(k)$:

$$A_+(k) = i\langle u_{+k}|\partial_k u_{+k}\rangle = i\left(\cos k/2 \quad -i\sin k/2\right)\begin{pmatrix}-1/2\,\sin k/2\\i/2\,\cos k/2\end{pmatrix} - 1/2 = -1/2. \tag{97}$$

Then, combining this with our expression for the Berry phase $\gamma_+$, we can show that in the tight-binding limit (see Exercise 10):

$$\gamma_+ = i\int_0^{2\pi}\mathrm{d}k\int_{cell}\mathrm{d}r\,u^*_{+k}(r)\partial_k u_{+k}(r) \tag{98}$$
$$= -\pi + i\int_0^{2\pi}\mathrm{d}k\int_{cell}\mathrm{d}r\,e^{-ikr}\left[\varphi_{sk}\cos\frac{k}{2} - i\varphi_{pk}\sin\frac{k}{2}\right]\left[\cos\frac{k}{2}\,\partial_k(e^{ikr}\varphi_{sk}) + i\sin\frac{k}{2}\,\partial_k(e^{ikr}\varphi_{pk})\right]$$
$$= -\pi,$$

and thus the center of charge for the corresponding Wannier function is $\gamma a/(2\pi) \bmod a = a/2$, where we have restored $a$ as lattice constant. (Hybrid) Wannier functions can be constructed exactly

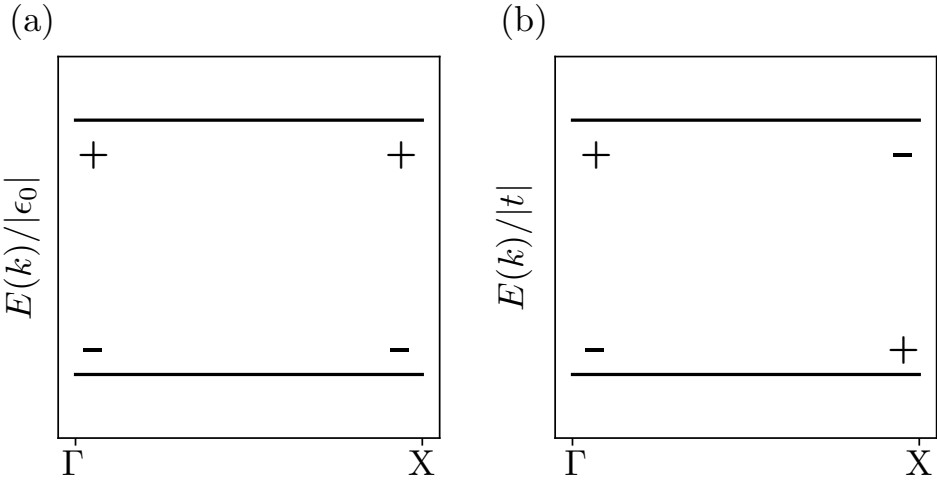

Figure 6: Flat bands of the Rice-Mele Hamiltonian in the two limits discussed in the text, with the inversion eigenvalues at $\Gamma$ and X labeled .

also in this case (See Exercise 11). Because inversion symmetry quantizes the polarization, we know that the Wannier centers are pinned at $a/2$ as long as the gap does not close. The localization length of the Wannier functions diverges as the gap is reduced.

To get a flavor of the next section, we can tie this result to the symmetry properties of the Bloch states at high symmetry points $\Gamma = (0)$ and $X = (\pi)$ in the Brillouin zone. Let us examine the symmetries of $\psi_{\pm k}(r)$ in both limits ① and ②. Recall that we took $U_I(k) = \sigma_z$. This yields

$$\text{①} \quad U_I \psi_{\pm\Gamma}(r) = U_I \varphi_{(s,p)\Gamma}(r) = \pm\psi_{\pm\Gamma}(r),$$
$$U_I \psi_{\pm X}(r) = U_I \varphi_{(s,p)X}(r) = \pm\psi_{\pm X}(r).$$

$$\text{②} \quad U_I \psi_{\pm\Gamma}(r) = U_I \varphi_{(s,p)\Gamma}(r) = \pm\psi_{\pm\Gamma}(r),$$
$$U_I \psi_{\pm X}(r) = U_I \varphi_{(p,s)X}(r) = \mp\psi_{\pm X}(r).$$

The flat bands obtained in both limits and the inversion eigenvalues of the corresponding eigenstates at $\Gamma$ and X are illustrated in Fig. 6.

Consulting the *Bilbao Crystallographic Server* [22], we find that the inversion eigenvalue distribution in ① matches what we would expect from orbitals at the $1a$ Wyckoff position transforming in the $A_g + A_u$ (s+p orbital) representation of inversion. We denote this as the $(A_g \uparrow G)_{1a} \oplus (A_u \uparrow G)_{1a}$ band representation–corresponding to s ($A_g$) and p ($A_u$) orbitals at the origin of the unit cell (the 1a position). Similarly, the inversion eigenvalues in ② match what we would expect for s and p orbitals at the 1$b$ Wyckoff position, which we denote as the $(A_g \uparrow G)_{1b} \oplus (A_u \uparrow G)_{1b}$ band representation corresponding to s ($A_g$) and p ($A_u$) orbitals half a lattice constant away from the origin of the unit cell (the 1b position). In fact, the construction of the Wannier functions in exercise 11 shows that these states are these band representations.

Note that we have accomplished something interesting in the transition from ①→②: by closing and reopening a gap, we have moved the centers of the Wannier functions from the atomic 1a position to the 1b position, half a unit cell away, generating a dipole moment of $ea/2$ per unit cell (recall that

$a$ is the lattice constant). The quantization of the dipole moment means that we could not have done this without either closing the gap or breaking inversion symmetry. The phases corresponding to the two limits are topologically distinct, but since both have exponentially localized Wannier functions, we refer to ② as an **obstructed atomic limit**. We will see in the next section that by breaking inversion and time-reversal symmetries, this is intimately related to topological insulators and the quantum Hall effect (QHE).

# 5 Topological Bands, Wilson Loops and Wannier Functions

In this section, we will define the Wilson loop and show how symmetries may constrain its spectrum. Then, we will learn that Wilson loop windings can be interpreted as an obstruction to constructing maximally localized Wannier functions and give an alternative interpretation in terms of the Chern number and gauge discontinuity. We will finish the section by exploring the obstruction in two models: the Thouless Pump and the Kane-Mele model.

## 5.1 Wilson Loops and Symmetries

In the previous sections, we have seen how adiabatic transport of Bloch functions reveals interesting information about the localization properties of (hybrid) Wannier states, as well as the geometry of projectors. Furthermore, we have seen in a particular model (Rice-Mele), how spatial symmetries like inversion can place constraints on the eigenvalues of $\mathcal{W}$, and hence on the position of charge centers corresponding to hybrid Wannier functions. We will now explore this relation more generally.

For the remainder of these notes, we will work with Bloch functions as if they were obtained from a tight-binding model. In such cases, as we have seen in the Rice-Mele chain, we start by constructing Bloch waves $\chi_\sigma^k(r)$ from a set of orthogonal tight-binding orbitals $\{\varphi_{\sigma R}(r)\}$ centered at $r_\sigma + R$:

$$\chi_{\sigma k}(r) = \frac{1}{\sqrt{N}} \sum_R e^{ik \cdot (R + r_\sigma)} \varphi_{\sigma R}(r), \tag{99}$$

where $\sigma$ denotes a collection of quantum numbers describing degrees of freedom within the unit cell such as position within the unit cell, orbital type, or spin. We can then expand eigenstates $\psi_{nk}(r)$ of the Hamiltonian as a linear combination of these Bloch waves as

$$\psi_{nk}(r) = \sum_\sigma u_{nk}^\sigma \chi_{\sigma k}(r) = \frac{1}{\sqrt{N}} \sum_{\sigma R} u_{nk}^\sigma e^{ik \cdot (R + r_\sigma)} \varphi_{\sigma R}(r). \tag{100}$$

Finally, the periodic part $u_{nk}(r)$ reads:

$$u_{nk}(r) = \sum_{\sigma R} u_{nk}^\sigma \varphi_{\sigma R}(r) e^{-ik \cdot (r - R - r_\sigma)}. \tag{101}$$

The periodicity of the eigenstates $\psi_{nk}(r)$ as $k \to k + G$ implies that

$$u_{nk+G}^\sigma = e^{-iG \cdot r_\sigma} \delta_{\sigma\sigma'} u_{nk}^{\sigma'} \equiv [V^{-1}(G)]_{\sigma\sigma'} u_{nk}^{\sigma'}. \tag{102}$$

Recall from Exercise 10 and Sec. 4.1 that Berry connections computed from $u_{nk}^\sigma$ and $u_{nk}(r)$ generally differ outside the tight-binding limit. Nevertheless, both connections obey the same symmetry constraints, because both $u_{nk}^\sigma$ and $u_{nk}(r)$ transform under (isomorphic) representations of the crystal

(a)

(b)

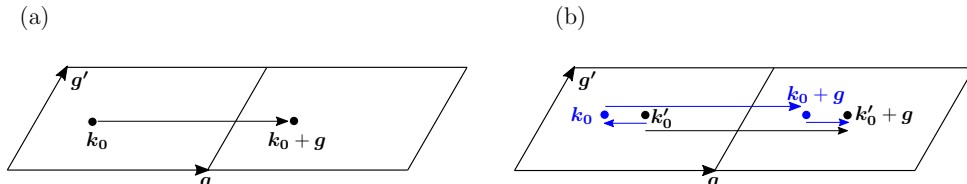

Figure 7: Paths in the BZ for a Wilson loop. (a) The simplest nontrivial closed path given in Eq. (108), winding once and parallel to a primitive reciprocal lattice vector $\boldsymbol{g}$. (b) A simple nontrivial path $\boldsymbol{k_0'} \to \boldsymbol{k_0'} + \boldsymbol{g}$ with basepoint $\boldsymbol{k_0'}$, and an alternative path with basepoint at $\boldsymbol{k_0}$.

symmetry group. Additionally, the geometry of adiabatic transport of the $u_{n\boldsymbol{k}}^\sigma$ is itself interesting, since they are eigenstates of the parametric family of matrix Hamiltonians

$$h_{\sigma\sigma'}(\boldsymbol{k}) = \int \mathrm{d}^d r \, \varphi_{\sigma\boldsymbol{k}}^*(\boldsymbol{r}) H(\boldsymbol{k}) \varphi_{\sigma'\boldsymbol{k}}(\boldsymbol{r}), \tag{103}$$

where we have (re)-introduced $\varphi_{\sigma\boldsymbol{k}}(\boldsymbol{r}) = \frac{1}{\sqrt{N}} \sum_{\boldsymbol{R}} \varphi_{\sigma\boldsymbol{R}}(\boldsymbol{r}) e^{-i\boldsymbol{k}\cdot(\boldsymbol{r}-\boldsymbol{R}-\boldsymbol{r}_\sigma)}$. In matrix notation, the Schrödinger equation for $u_{n\mathbf{k}}^\sigma$ becomes

$$h(\boldsymbol{k}) \begin{pmatrix} u_{n\boldsymbol{k}}^1 \\ u_{n\boldsymbol{k}}^2 \\ \vdots \end{pmatrix} = E_{n\boldsymbol{k}} \begin{pmatrix} u_{n\boldsymbol{k}}^1 \\ u_{n\boldsymbol{k}}^2 \\ \vdots \end{pmatrix}. \tag{104}$$

For the rest of these notes we will focus on the parallel transport of the projectors $[P(\boldsymbol{k})]_{\sigma\sigma'} = \sum_{n=1}^N u_{n\boldsymbol{k}}^{\sigma*} u_{n\boldsymbol{k}}^{\sigma'}$. We will denote $|u_{n\boldsymbol{k}}\rangle$ the column vector of coefficients $u_{n\boldsymbol{k}}^\sigma$, so that $P(\boldsymbol{k}) = \sum_{n=1}^N |u_{n\boldsymbol{k}}\rangle\langle u_{n\boldsymbol{k}}|$. Let us consider the holonomy matrix $W_{\mathcal{C}}^{nm}$ along a smooth contour $\mathcal{C}$ in the BZ given by

$$W_{\mathcal{C}}^{nm} = \left\langle u_{n\boldsymbol{k}_f} \middle| \mathcal{W}_{\mathcal{C}} \middle| u_{m\boldsymbol{k}_0} \right\rangle = \left\langle u_{n\boldsymbol{k}_f} \middle| \prod_{\boldsymbol{k}}^{\mathcal{C}} P(\boldsymbol{k}) \middle| u_{m\boldsymbol{k}_0} \right\rangle, \tag{105}$$

where $\mathcal{C}$ starts at $\boldsymbol{k}_0$ and ends at $\boldsymbol{k}_f$. By construction, each projector is invariant under a $U(N)$-valued gauge transformations $U(\boldsymbol{k})$ at each $\boldsymbol{k}$,

$$U(\boldsymbol{k}) P(\boldsymbol{k}) U^\dagger(\boldsymbol{k}) = P(\boldsymbol{k}). \tag{106}$$

As such, by defining $|u_{n\boldsymbol{k}}'\rangle = U_{nm}^\dagger(\boldsymbol{k})|u_{m\boldsymbol{k}}\rangle$, the holonomy matrix $W_{\mathcal{C}}$ transforms into $W_{\mathcal{C}}'$ in the following way:

$$W_{\mathcal{C}}' = U^\dagger(\boldsymbol{k}_f) W_{\mathcal{C}} U(\boldsymbol{k}_0), \tag{107}$$

thus, like all adiabatic transport, the spectrum of $W_{\mathcal{C}}$ is gauge invariant only when $\mathcal{C}$ is a closed curve. The holonomy $\mathcal{W}_{\mathcal{C}}$ for a closed loop $\mathcal{C}$ is referred to as **Wilson Loop**. For simple (i.e. contractible) closed curves, this is the end of the story. However, recall that the Brillouin Zone is topologically a $d$-dimensional torus. Thus, there are nontrivial cycles $\mathcal{C}_{\boldsymbol{g}}$ which wind from $\boldsymbol{k}_0$ to $\boldsymbol{k}_0 + \boldsymbol{g}$, with $\boldsymbol{g}$ a reciprocal lattice vector. The simplest such curves are linear and wind only once, with $\boldsymbol{g} = \boldsymbol{G}$ a primitive reciprocal lattice vector as sketched in Fig. 7a and given analytically by

$$\mathcal{C}_{\boldsymbol{g}} = \{\boldsymbol{k_0} + \boldsymbol{g} \; t \mid t \in [0,1]\}. \tag{108}$$

Recall also from Sec. 4 that eigenvalues of $\mathcal{W}_{\mathcal{C}_g}$ give the charge centers of hybrid Wannier functions that are exponentially localized in the direct lattice direction $a$ that is not orthogonal to $\boldsymbol{g}$. Since periodicity with respect to translations of the reciprocal lattice requires that $|u_{nk+g}\rangle = V^{-1}(\boldsymbol{g})|u_{nk}\rangle$, we must be careful to ensure that $\mathcal{W}_{\mathcal{C}_g}$ is closed in a way that obeys this boundary condition, implying

$$W_{\mathcal{C}_g}^{nm} = \langle u_{nk+g}|\mathcal{W}_{\mathcal{C}_g}|u_{mk}\rangle = \langle u_{nk}|V(\boldsymbol{g})\mathcal{W}_{\mathcal{C}_g}|u_{mk}\rangle. \tag{109}$$

This means that the operator $V(\boldsymbol{g})\mathcal{W}_{\mathcal{C}_g}$ describes parallel transport along the closed non-contractible cycle $\mathcal{C}_{\boldsymbol{g}}$. We thus have that the Wilson loop can be expressed as

$$\mathcal{W}_{\boldsymbol{g},\boldsymbol{k_0}} = V(\boldsymbol{g})\mathcal{W}_{\mathcal{C}_g} = V(\boldsymbol{g}) \prod_{k}^{\boldsymbol{k_0}+\boldsymbol{g}\leftarrow\boldsymbol{k_0}} P(\boldsymbol{k}), \tag{110}$$

whose nonzero eigenvalues are gauge invariant and correspond, in the tight-binding limit, to the centers of hybrid Wannier functions localized in the in the $\boldsymbol{r}\cdot\hat{\boldsymbol{g}}$ direction. But what is the role of the basepoint $\boldsymbol{k_0}$? Consider the paths $\boldsymbol{k_0'} \to \boldsymbol{k_0'} + \boldsymbol{g}$ and $\boldsymbol{k_0} \to \boldsymbol{k_0} + \boldsymbol{g}$, shown in Fig. 7b, which have basepoints that are shifted in the $\hat{\boldsymbol{g}}$-direction By making use of the expression for the Wilson loop as a product of projectors, combined with unitarity, we deduce that[7]

$$W_{\boldsymbol{g},\boldsymbol{k_0'}}^{mn} = W_{\boldsymbol{k_0'}+\boldsymbol{g}\leftarrow\boldsymbol{k_0}+\boldsymbol{g}}^{ml} \; W_{\boldsymbol{g},\boldsymbol{k_0}}^{lp} \; W_{\boldsymbol{k_0}\leftarrow\boldsymbol{k_0'}}^{pn} = [W_{\boldsymbol{k_0}\leftarrow\boldsymbol{k_0'}}^{\dagger}]^{ml} \; W_{\boldsymbol{g},\boldsymbol{k_0}}^{lp} \; W_{\boldsymbol{k_0}\leftarrow\boldsymbol{k_0'}}^{pn}. \tag{111}$$

This means that Wilson loops $W_{\boldsymbol{g}}$ starting from basepoints that differ in the $\hat{\boldsymbol{g}}$ direction are related by a similarity transformation. Thus although Wilson loop matrices with different basepoints have different matrix elements, they share the same spectrum; hybrid Wannier centers do not depend on the choice of the basepoint of the Wilson loop. Based on this observation, we will omit the basepoint $\boldsymbol{k_0}$ of the loop for brevity.

Additionally, when we face systems defined in 2 or 3 dimensions, the Wilson loop matrix will depend on the component of $\boldsymbol{k}$ perpendicular to the direction along which the loop runs. If we decompose $\boldsymbol{k}$ into parallel $k_\parallel$ and perpendicular $\boldsymbol{k_\perp}$ components, such that $\boldsymbol{k} = (k_\parallel, \boldsymbol{k_\perp})$, then we can write $W_{\boldsymbol{g}} = W_{\boldsymbol{g}}(\boldsymbol{k_\perp})$.

Now, let us focus on the action of space group symmetries. A space group symmetry operation $s = \{R|\boldsymbol{v}\}$ acts on the vector of coefficients $\{u_{nk}^\sigma\}$ as

$$u_{nk}^\sigma \to U_R^{\sigma\sigma'} u_{n(Rk)}^{\sigma'} e^{-i(Rk)\cdot\boldsymbol{v}} \equiv S_k^{\sigma\sigma'} u_{n(Rk)}^{\sigma'}. \tag{112}$$

Let us examine two important cases:

① **Inversion symmetry $\{I|\boldsymbol{0}\}$:**

According to Eq. (112), under inversion the projector $P(\boldsymbol{k})$ transforms as

$$U_I P(k) U_I^\dagger = P(-k). \tag{113}$$

We want to study the way in which the Wilson loop operator $\mathcal{W}_{\boldsymbol{g}}(\boldsymbol{k_\perp})$ transforms under inversion. Writing out the product of projectors, we have

$$U_I \mathcal{W}_{\boldsymbol{g}}(\boldsymbol{k_\perp}) U_I^\dagger = \lim_{\delta\to 0} U_I V(\boldsymbol{g}) P(\boldsymbol{g},\boldsymbol{k_\perp}) P(\boldsymbol{g}-\boldsymbol{\delta},\boldsymbol{k_\perp})\dots P(0) U_I^\dagger. \tag{114}$$

---

[7]Remember that $W$ is the matrix of $\mathcal{W}$ restricted to the subspace of the image of projectors.

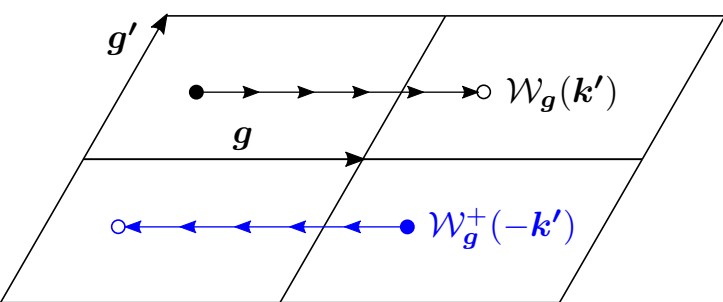

Figure 8: In black, the graphical discretization of the Wilson loop $\mathcal{W}_g(\boldsymbol{k}')$ applied in the derivation of the effect of inversion and time-reversal on it. In blue, the Wilson loop $\mathcal{W}_g^\dagger(-\boldsymbol{k}')$ to which it is related by these symmetries.

Having in mind that $U_I$ is unitary, we insert the identity $U_I^\dagger U_I$ between $V(\boldsymbol{g})$ and $P(\boldsymbol{g},\boldsymbol{k}_\perp)$, and also between every pair of projectors and apply Eq. (113) to find

$$U_I \mathcal{W}_g(\boldsymbol{k}_\perp) U_I^\dagger = \lim_{\delta \to 0} U_I V(\boldsymbol{g}) U_I^\dagger P(-\boldsymbol{g},-\boldsymbol{k}_\perp) P(-\boldsymbol{g}+\boldsymbol{\delta},-\boldsymbol{k}_\perp) P(-\boldsymbol{g}+2\boldsymbol{\delta},-\boldsymbol{k}_\perp)\ldots P(0). \quad (115)$$

We need to work out the relation between $U_I$ and $V(\boldsymbol{g})$. On the one hand

$$P(\boldsymbol{g},\boldsymbol{k}_\perp) = V^\dagger(\boldsymbol{g})P(0,\boldsymbol{k}_\perp)V(\boldsymbol{g}) = V^\dagger(\boldsymbol{g})U_I P(0,-\boldsymbol{k}_\perp)U_I^\dagger V(\boldsymbol{g}), \quad (116)$$

on the other hand

$$P(\boldsymbol{g},\boldsymbol{k}_\perp) = U_I P(-\boldsymbol{g},-\boldsymbol{k}_\perp)U_I^\dagger = U_I V(\boldsymbol{g})P(0,-\boldsymbol{k}_\perp)V^\dagger(\boldsymbol{g})U_I^\dagger. \quad (117)$$

To be consistent, we must have

$$U_I V(\boldsymbol{g}) = V^\dagger(\boldsymbol{g})U_I. \quad (118)$$

Applying this and the fact that $V^\dagger(\boldsymbol{g}) = V(-\boldsymbol{g})$ in Eq. (115), we see that

$$\begin{aligned}
U_I \mathcal{W}_g(\boldsymbol{k}_\perp) U_I^\dagger &= \lim_{\delta \to 0} V(-\boldsymbol{g})P(-\boldsymbol{g},-\boldsymbol{k}_\perp)P(-\boldsymbol{g}+\boldsymbol{\delta},-\boldsymbol{k}_\perp)P(-\boldsymbol{g}+2\boldsymbol{\delta},-\boldsymbol{k}_\perp)\ldots P(0) \\
&= W_g^\dagger(-\boldsymbol{k}_\perp).
\end{aligned} \quad (119)$$

In conclusion, $\mathcal{W}_g(\boldsymbol{k}_\perp)$ and $\mathcal{W}_g^\dagger(-\boldsymbol{k}_\perp)$ are isospectral. In particular, for inversion-invariant momenta $\boldsymbol{k}_\perp \equiv -\boldsymbol{k}_\perp$ (where $\equiv$ denotes equivalence modulo a reciprocal lattice vector), this implies that $\mathcal{W}_g(\boldsymbol{k}_\perp)$ and $\mathcal{W}_g^\dagger(\boldsymbol{k}_\perp)$ are isospectral, so that eigenvalues of $\mathcal{W}_g(\boldsymbol{k}_\perp)$ are either real, or come in complex conjugate pairs (See Exercise 9).

② **Time-reversal symmetry $\mathcal{T} = U_T \mathcal{K}$:**

Let us consider the action of time-reversal symmetry on the Wilson loop operator:

$$\begin{aligned}
\mathcal{T} \mathcal{W}_g(\boldsymbol{k}_\perp) \mathcal{T}^{-1} &= \mathcal{T} V(\boldsymbol{g}) \prod^{g \leftarrow 0} P(\boldsymbol{k}_\perp) \mathcal{T}^{-1} = V(-\boldsymbol{g}) \prod^{g \leftarrow 0} \mathcal{T} P(\boldsymbol{k}_\perp) \mathcal{T}^{-1} \\
&= V(-\boldsymbol{g}) \prod^{-g \leftarrow 0} P(-\boldsymbol{k}_\perp) = \mathcal{W}_g^\dagger(-\boldsymbol{k}_\perp).
\end{aligned} \quad (120)$$

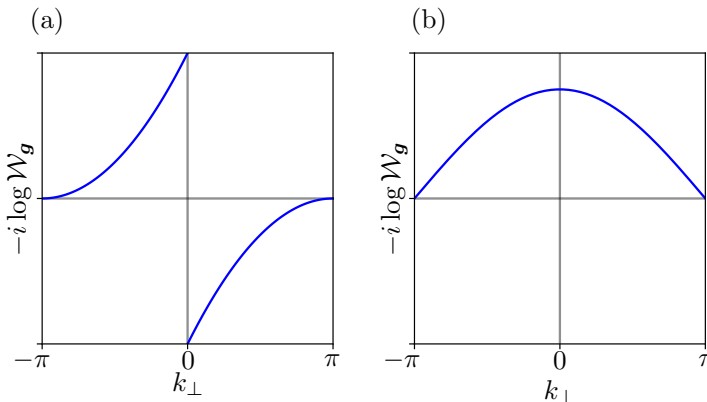

Figure 9: Example Wilson loop spectra of systems with (a) inversion symmetry (b) time-reversal symmetry.

It is left as an exercise (Exercise 13) to show that this relation leads to the conclusion that $\mathcal{W}_g(\boldsymbol{k}_\perp)$ and $\mathcal{W}_g(-\boldsymbol{k}_\perp)$ are isospectral. What is more, for spinful electrons ($\mathcal{T}^2 = -1$), the spectra of the Wilson loop operator $W_g(\boldsymbol{k}_\perp)$ has a Kramers degeneracy if $-\boldsymbol{k}_\perp \equiv \boldsymbol{k}_\perp$, so that each eigenvalues at time-reversal invariant momenta (TRIMs) are doubly degenerate.

A similar analysis can be carried out for any symmetry operation (see, e.g. Ref. [23]), which may be helpful to investigate the constraints that symmetries place on Wilson loop spectra in more complicated space groups. Nevertheless, we will only make use of time reversal and inversion symmetry in what follows.

To contextualize these results, let us return to the 1D Rice-Mele chain. According to our analysis, inversion symmetry forces eigenvalues of $\mathcal{W}_g$ to be real or come in complex conjugate pairs. Since there exists a single occupied band, the nonzero eigenvalue $\lambda$ of $\mathcal{W}_g$ should be real, and so it must be either 1 or -1; equivalently: $(2i\pi)^{-1}\log\lambda = 0, 1/2$. Now, we have a wider picture of how inversion symmetry quantizes hybrid Wannier centers in 1D. In systems defined in more dimensions, we do not have this quantization for generic $\boldsymbol{k}_\perp$, because $\boldsymbol{k}_\perp \not\equiv -\boldsymbol{k}_\perp$ in general. However, we gain something amazing: the possibility of finding **topologically distinct** spectra for $\mathcal{W}_g(\boldsymbol{k}_\perp)$ as a function of $\boldsymbol{k}_\perp$.

## 5.2 Wilson Loop Winding and Wannier Obstruction

To motivate this discussion, let us recall that in 1D, the notions of Wannier and hybrid Wannier functions coincide. Thus, in the tight-binding limit in 1D, the **Wannier centers** coincide with $(2\pi i)^{-1}$ times the logarithm of eigenvalues of the Wilson loop $\mathcal{W}_g$ (mod $a$). In higher dimensions, this is not generically the case even in the tight-binding limit, because projected position operators along different directions need not commute,

$$[Px_iP, Px_jP] \neq 0. \tag{121}$$

In such cases, it is not possible to simultaneously diagonalize all the projected position operators. In other words, generally it is not possible to find functions that are simultaneous eigenstates of

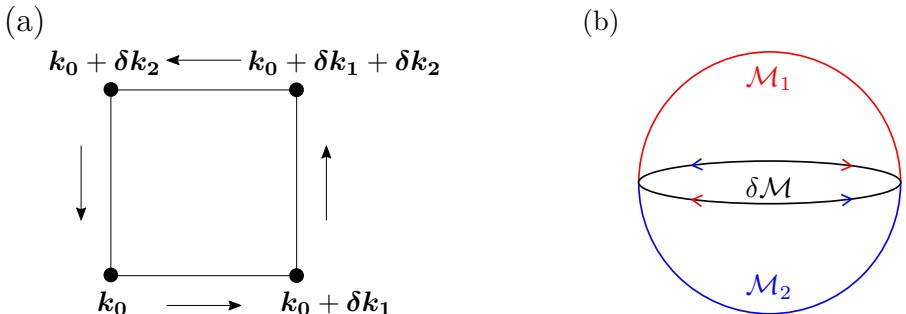

Figure 10: (a) Path considered in the statement of Ambrose-Singer theorem in Eq. (124). (b) Sphere $\mathcal{M} = \mathcal{M}_1 \cup \mathcal{M}_2$ considered as closed manifold for the definition of the (first) Chern number.

projected positions along multiple directions. To see how this can happen, let us take a trial state $|f\rangle = \sum_{n,\boldsymbol{k}} f_{n\boldsymbol{k}} |\psi_{n\boldsymbol{k}}\rangle \in \mathrm{Im}(P)$. Using Eq. (76), we have

$$
\begin{aligned}
[Px_iP, Px_jP]|f\rangle &= \sum_{n=1}^{N} \sum_{\boldsymbol{k},mnl} \left( i\partial_i A_{nm}^j - i\partial_j A_{nm}^i + A_{nl}^i A_{lm}^j - A_{nl}^j A_{lm}^i \right) f_{m\boldsymbol{k}} |\psi_{n\boldsymbol{k}}\rangle \\
&= i \sum_{n=1}^{N} \sum_{\boldsymbol{k},mn} \Omega_{nm}^{ij}(\boldsymbol{k}) |\psi_{n\boldsymbol{k}}\rangle f_{m\boldsymbol{k}},
\end{aligned}
\tag{122}
$$

where

$$
\Omega_{nm}^{ij}(\boldsymbol{k}) = \partial_i A_{nm}^j(\boldsymbol{k}) - \partial_j A_{nm}^i(\boldsymbol{k}) - i[A^i(\boldsymbol{k}), A^j(\boldsymbol{k})]_{nm}
\tag{123}
$$

is the **Berry-curvature** tensor. We see that, in order for there to exist a basis in which $Px_iP$ and $Px_jP$ are simultaneously diagonal, the Berry curvature tensor should vanish for all $\boldsymbol{k}$. This means, generically, that hybrid Wannier functions–eigenstates of a single $Px_iP$–will not coincide with maximally localized Wannier functions–orbitals designed to be as localized as possible in all directions. Thus, we must take care to distinguish between hybrid Wannier centers and Wannier centers. This is particularly important because, while unique hybrid Wannier functions exist for any gapped projector (they are eigenstates of the projected position operator, $Px_iP$), Wannier functions are not unique and may not even be exponentially localizable (while respecting symmetries).

Eq. (122) and the fact that the Wilson loop is related to the Berry connection suggest that information about the (tight-binding) Berry curvature is contained in the Wilson loop. In order to get a deeper insight into this relation, we need to introduce the Ambrose-Singer theorem, which relates the holonomy of a connection to its curvature. Let us consider a parallelogram in the Brillouin Zone with sides of infinitesimal length (See Fig. 10a). The **Ambrose-Singer theorem** gives the leading order term in the Taylor series of the Wilson loop along the boundary of the parallelogram in terms of the Berry curvature via

$$
i\Omega_{nm}^{12}(k_0)\delta k_1 \delta k_2 = \log W_4 W_3 W_2 W_1 + \mathcal{O}(\delta k^3).
\tag{124}
$$

This means that for infinitesimally small square paths, the leading contribution to the Wilson loop is given by the Berry curvature. This relation between Wilson loops and Berry curvature may be more

familiar when we take the trace of both sides. Upon taking the trace of the right hand side, we see that

$$\mathrm{tr}\log W_4 W_3 W_2 W_1 = \log\det\{W_4 W_3 W_2 W_1\}. \tag{125}$$

Since the product of a determinant of matrices equals the determinant of their product, taking the trace removes any concern about noncommutativity of the $W_i$. We can then use Stokes's theorem to go beyond infinitesimal parallelograms, and consider instead paths that enclose finite regions of the Brillouin zone. To be precise, let us look at the trace of the left hand side of Eq. (124). Terms coming from the matrix multiplication of $A^i(k)$ and $A^j(k)$ in Eq. (123) do not contribute to the trace due to the cyclic property, $\mathrm{tr}([X,Y]) = \mathrm{tr}(XY) - \mathrm{tr}(YX) = 0$. We can then add up a series of infinitesimal Wilson loops to create a finite region, as in Fig. 11b. Then, from Stokes's theorem[8], we have that:

$$\frac{1}{2\pi}\int_M \mathrm{tr}(\Omega^{12})\,dk^1\,dk^2 = \frac{1}{2\pi}\int_M (\partial_1 \mathrm{tr}A^2 - \partial_2 \mathrm{tr}A^1)\,dk^1\,dk^2 = \frac{1}{2\pi}\oint_{\partial M} \mathrm{tr}A \cdot d\boldsymbol{l} \tag{126}$$

$$= \frac{1}{2\pi i}\int_M d\boldsymbol{k}\,\log\det W_1 W_2 W_3 W_4\,, \tag{127}$$

where the last equality comes from using the Ambrose-Singer theorem and superscripts denote directions in reciprocal space. If $\mathcal{M}$ is a closed manifold (such as a plane "bounded" by reciprocal lattice vectors $\boldsymbol{g}_1$ and $\boldsymbol{g}_2$ in the 2D Brillouin Zone), this integral vanishes, modulo gauge discontinuities in $\mathrm{tr}A$. As an example, consider the sphere $\mathcal{M}$ of Fig. 10b. We divide the sphere into upper and lower patches $\mathcal{M}_1$ and $\mathcal{M}_2$, respectively. At the equator, wave functions defined in top and bottom patches must be equal up to a gauge transformation

$$|\psi_{n\boldsymbol{k}}^{\mathcal{M}_2}\rangle = U_{nm}(\boldsymbol{k})|\psi_{m\boldsymbol{k}}^{\mathcal{M}_1}\rangle\,, \tag{128}$$

where $U$ is a unitary matrix. This implies that, at the equator, the Berry connections $\boldsymbol{A}_1$ and $\boldsymbol{A}_2$ in the two patches are related via

$$\boldsymbol{A}_2 = U^\dagger \boldsymbol{A}_1 U + i U^\dagger \boldsymbol{\nabla}U\,, \tag{129}$$

which implies that

$$\mathrm{tr}\boldsymbol{A}_2 = \mathrm{tr}\boldsymbol{A}_1 + i\,\mathrm{tr}(U^\dagger\boldsymbol{\nabla}U) = \mathrm{tr}\boldsymbol{A}_1 - \boldsymbol{\nabla}\varphi\,, \tag{130}$$

where we have defined $\varphi = \mathrm{Im}\log\det U$ as the sum of the phase of the eigenvalues of $U$. With this in mind, we find that the integral of the Berry curvature over the sphere is

$$\frac{1}{2\pi}\int_M \mathrm{tr}(\Omega)\,d^2k = \frac{1}{2\pi}\int_{M_1} \mathrm{tr}(\Omega)\,d^2k + \frac{1}{2\pi}\int_{M_2} \mathrm{tr}(\Omega)\,d^2k$$

$$= \frac{1}{2\pi}\int_{\partial M} \mathrm{tr}(\boldsymbol{A}_1)\cdot d\boldsymbol{l} - \frac{1}{2\pi}\int_{\partial M} \mathrm{tr}(\boldsymbol{A}_2)\cdot d\boldsymbol{l} \tag{131}$$

$$= \frac{1}{2\pi}\int_{\partial M} \boldsymbol{\nabla}\varphi\cdot d\boldsymbol{l}\,.$$

---

[8]Note that, when the subspace of interest contains more than one band, Ambrose-Singer theorem may not be equivalent to Stokes's theorem (See Exercise 14).

By periodicity of the gauge transformation $U$, this integral must be an integer $\nu$, called first **Chern number**

$$\frac{1}{2\pi} \int_M \text{tr}(\Omega) \cdot \text{d}^2 k = \nu \in \mathbb{Z}, \tag{132}$$

which is a topological invariant of states defined on the closed manifold $\mathcal{M}$.

Returning to the Brillouin zone, we have figured out how the Chern number arises in the left-hand side of the Ambrose-Singer theorem, Eq. (124) by taking a trace and integrating over the whole BZ. If we also work with the right-hand side of (124), we can relate the Chern number to the spectrum of the Wilson loop. As an example, let us consider a 2D plane $\{(k_1 \boldsymbol{g}_1, k_2 \boldsymbol{g}_2) \mid k_1, k_2 \in [0, 1]\}$ in the BZ and let us compare $\mathcal{W}_{\boldsymbol{g}_2}(k_1)$ and $\mathcal{W}_{\boldsymbol{g}_2}(k_1 + \Delta k)$:

$$\log \det \left[ W_{\boldsymbol{g}_2}(k_1 + \Delta k) W_{\boldsymbol{g}_2}^{\dagger}(k_1) \right] = \log \det \left( \square \square \ldots \square \right) = i \int_M \text{tr}(\Omega) \, \text{d}^2 k + \mathcal{O}(\Delta k^2), \tag{133}$$

where $M$ is the region between the two loops, and $\square \square \ldots \square$ schematically represents dividing region $M$ into Wilson loops evaluated on plaquettes, as shown in Fig. 11b. Note that we have made use of the fact that $W_{\boldsymbol{g}_2}$ is evaluated on a loop traversing the BZ which, together with the fact that we are considering the determinant, allows us to neglect the initial and final horizontal segments in Fig.11b. In the limit $\Delta k \to 0$, we see that $\text{tr}(\Omega)$ controls the change in $\log \det(W_{\boldsymbol{g}_2})$. In particular,

$$\partial_{k_1} \log \det W_{\boldsymbol{g}_2}(k_1) = i \int \text{d}k_2 \, \Omega^{12}(\boldsymbol{k}), \tag{134}$$

and so

$$\frac{1}{2\pi i} \int \text{d}k_1 \, \partial_{k_1} \log \det W_{g_2}(k_1) = \nu, \tag{135}$$

which can be obtained by neglecting $\mathcal{O}(\Delta k^2)$ terms in the Taylor expansion of the left hand side of Eq. (133). The last integral in Eq. (134) is the number of times the sum of hybrid Wannier centers winds across the entire unit cell and $\nu$ is the Chern number. For example, in the case of Fig. 11a, we have 2 centers winding upwards, 1 downwards, so in total $\nu = 2 - 1 = 1$.

The Chern number defines classes of insulating Hamiltonians which cannot be deformed into each other without closing a gap, since

a) $\oint \text{tr}\,\Omega \, \text{d}^2 k$, being an integer, cannot change under small perturbations of the Hamiltonian.

b) Periodicity of both the hybrid Wannier centers and the Brillouin zone implies that eigenvalues of the Wilson loop cannot smoothly unwind.

This means that projectors with different values of $\nu$ are **topologically distinct**. Even more radically, if $\nu \neq 0$, it is not possible to construct exponentially localized Wannier functions for the projectors, as there fails to exist a smooth gauge that allows us to construct Bloch waves $\tilde{\psi}_{n\boldsymbol{k}}(\boldsymbol{r})$ satisfying Eq. (69). To see this, recall how we originally defined the Chern number for the sphere: We showed that $\boldsymbol{A}_1$ and $\boldsymbol{A}_2$ (the Berry connections in either patch of the sphere) are related by a gauge transformation at the intersection of the two patches, and that this gauge transformation has a nontrivial winding number equal to the Chern number. But a unitary matrix, like the gauge transformation $U(\boldsymbol{k})$, cannot wind if it is globally defined (imagine shrinking one of the patches to a

(a)                     (b)

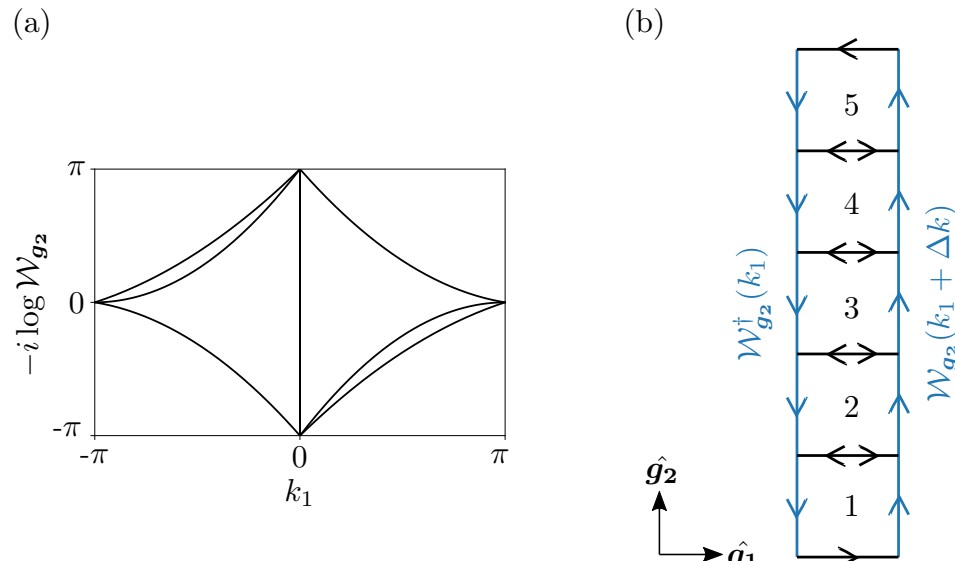

Figure 11: a) Wilson loop spectrum for a system with Chern number $\nu = 1$. b) In blue, Wilson loops along $\boldsymbol{g}_2$ at $k_1$ and $k_1 + \Delta k$. Taking the logarithm of the determinant in Eq. (133) allows us to close the path in the border of the BZ and apply Stokes theorem, to relate the Wilson loop spectrum to the Chern number. Horizontal segments do not contribute to the integral.

point). Thus, the Chern number can only be nonzero when there fails to exist a global smooth gauge choice for the wavefunctions. This means that the Bloch vectors $u_{n\boldsymbol{k}}^{\sigma}$ **must**[9] have a phase singularity somewhere in the BZ. In particular, any globally defined gauge must have a singularity in some point of the BZ, so following our discussion of Sec. 4, Wannier functions associated to these bands cannot be exponentially localized. Thus, we have the important result that

> The Chern Number is a Wannier Obstruction.

Alternatively, we can interpret the obstruction pictorially from our Wilson loop formulation of the Chern number, by noting that if an eigenvalue $e^{i\theta_n}$ of the Wilson loop winds $\nu$ times, then the hybrid Wannier functions $|W_{n\boldsymbol{R}_2}(\boldsymbol{k}_1)\rangle$ and $|W_{n\boldsymbol{R}_2}(\boldsymbol{k}_1 + \boldsymbol{g}_1)\rangle$ have centers of charge which differ by $\nu$ unit cells. Thus, the hybrid Wannier functions are not periodic in $\boldsymbol{k}_1$ and cannot be Fourier transformed to get exponentially localized Wannier functions.

By means of this picture of the Chern number as a "pump" of hybrid Wannier function centers, we can construct an example of a "Chern insulator" based on the Rice-Mele chain, namely *the Thouless Pump*.

## 5.3 The Thouless Pump

In this section, we will present a model for a topological insulator with a non-vanishing Chern number. Although we will introduce it as an extension of the Rice-Mele model, it can be understood as a 2D system with broken time-reversal (TR) symmetry.

---

[9]$u_{n\boldsymbol{k}} = U_{nm}(\boldsymbol{k})u_{m\boldsymbol{k}}$ and $U(\boldsymbol{k})$ cannot be globally extended to the whole BZ.

Recall our simplified tight-binding Hamiltonian for the Rice-Mele chain,

$$h(k_1) = (\epsilon + t\cos k_1)\sigma_z + t\sin k_1 \sigma_y\,, \tag{136}$$

where we have renamed $k \to k_1$. We showed in Sec. 4.1 that when $t < \epsilon$, the eigenvalue of the Wilson loop for the valence band is 1, while for $t > \epsilon$ it is $-1$. Let us imagine that $\epsilon$ and $t$ depend on a periodic parameter denoted $k_2$, which goes from $-\pi \to \pi$ and is odd under inversion and TR symmetry. We can then rewrite the Hamiltonian as $h(k_1, k_2)$. If we can ensure that inversion symmetry is preserved and that $h(k_1, 0)$ has $W_{g_1} = 1$ while $h(k_1, \pi)$ has $W_{g_1} = -1$, then we will have a model which, at a minimum, pumps the (hybrid) Wannier centers from $R_1 = 0$ to $R_1 = 1$ as a function of $k_2$; such a model would have[10] $\nu = -1$. Note that this requires breaking TR symmetry, since TR symmetry forces the Wilson loop matrices $W_{g_1}(k_2)$ and $W_{g_1}(-k_2)$ to be isospectral[11]. To satisfy these requirements, we can take

$$h(k_1, k_2) = a\big[(1 + \cos k_1 + \cos k_2)\sigma_z + \sin k_1 \sigma_y + \sin k_2 \sigma_x\big]\,, \tag{137}$$

which at $k_2 = 0$ and $k_2 = \pi$ becomes

$$\begin{aligned} h(k_1, 0) &= a\big[(2 + \cos k_1)\sigma_z + \sin k_1 \sigma_y\big]\,, \\ h(k_1, \pi) &= a\big[\cos k_1 \sigma_z + \sin k_1 \sigma_y\big]\,. \end{aligned} \tag{138}$$

We see that as a function of $k_2$, the Hamiltonian $h(k_1, k_2)$ interpolates between a 1D inversion symmetric chain (Rice-Mele chain) Hamiltonian with valence band inversion ($\sigma_z$) eigenvalues[12] ($--$) at $k_2 = 0$ and one with valence band inversion eigenvalues ($-+$) at $k_2 = \pi$. In terms of the eigenvalue of the Wilson loop for the valence band this implies that

$$\begin{aligned} W_1(k_2 = 0) &= +1\,, \\ W_1(k_2 = \pi) &= -1\,. \end{aligned} \tag{139}$$

Thus, $\mathrm{Im}\log W_{g_1}(k_2)$ has the spectrum shown in Fig. 12, which corresponds to the Chern number $\nu = -1$. According to our discussion, this indicates that there is an obstruction to constructing exponentially localized Wannier functions, and a topological distinction between projectors.

Note that the $\sigma_x$ term in $h(k_1, k_2)$ plays two important roles. First, it ensures the existence of a gap for all $k_1$ and $k_2$. Second, as we mentioned, it breaks TR symmetry and allows $W_{g_1}(k_2)$ and $W_{g_1}(-k_2)$ to have different spectra, thus allowing for the winding in the Wilson loop spectrum.

Two comments about this model are in order:

1. While inversion symmetry simplifies the analysis by pinning $W_1(0, \pi)$ to $\pm 1$, it is not necessary to define the Chern number. The winding of the Wilson loop spectrum–and hence the Chern number–is robust to inversion symmetry breaking.

2. Recall from Sec. 3 that a small electric field applied in the $\boldsymbol{R}_2$ direction will adiabatically shift $k_2$. From Fig. 12 , we see that this will adiabatically shift the hybrid Wannier centers in the $\boldsymbol{R}_1$ direction, generating a current. Therefore, $\nu$ governs the quantization of the **Hall conductance**.

---

[10]assuming $\boldsymbol{g}_1$ and $\boldsymbol{g}_2$ form a right-handed coordinate system

[11]We can also prove the Bulk Boundary Correspondence: the spectrum of $PxP$ can be deformed to the spectrum of surface potential $\theta(x - x_0)$. See, e.g., Refs. [24, 25]

[12]The first sign corresponds to the high-symmetry point $\Gamma$, while the second sign to $X$.

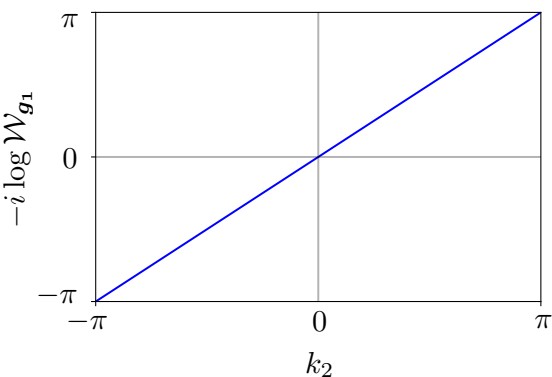

Figure 12: Wilson loop eigenvalue of the valence band in the model for the Thouless pump. Notice that, as was explained in detail in Sec. 5.1, inversion symmetry requires the spectrum to be antisymmetric about $k_2 = 0$.

We have seen that when $\det(W)$ winds, i.e. when $\nu \neq 0$, there is an obstruction to constructing exponentially localized Wannier functions, and hence also a topological distinction between projectors. In the presence of additional symmetries, we can generalize this significantly by looking at the entire spectrum of the Wilson loop rather than just its determinant. As we saw earlier, symmetries may protect degeneracies in the Wilson loop spectrum. When this happens, individual Wilson loop eigenvalues may wind, even if the determinant of the Wilson loop is trivial ($\nu = 0$). Then, adiabatic deformations that preserve symmetries cannot deform the spectrum of $W$ to a spectrum consistent with any atomic limit. These *topological crystalline phases* include concepts such as mirror Chern insulators (Wilson loop eigenvalue crossings protected by mirror symmetry eigenvalues, as in Exercise 12) and TR-invariant topological insulators (Wilson crossings protected by Kramers theorem, as in Exercise 13). To conclude, we will examine the simplest example of the latter, by means of the **Kane-Mele** model.

## 5.4 Kane-Mele Model

Let us consider a model that consists of $p_z$ orbitals sitting on a honeycomb lattice, whose symmetry group is the layer group p6/mmm (isomorphic to space group 191 when we forget about translations in the $z$-direction). We choose as a basis for the Bravais lattice the vectors

$$
\begin{aligned}
\mathbf{e}_1 &= \frac{1}{2}(\sqrt{3}, -1), \\
\mathbf{e}_2 &= \frac{1}{2}(\sqrt{3}, 1).
\end{aligned}
\tag{140}
$$

In this basis, the honeycomb lattice sites are given (within the unit cell) by

$$
\begin{aligned}
\mathbf{q}_A &= \frac{1}{3}\mathbf{e}_1 + \frac{1}{3}\mathbf{e}_2, \\
\mathbf{q}_B &= \frac{2}{3}\mathbf{e}_1 + \frac{2}{3}\mathbf{e}_2.
\end{aligned}
\tag{141}
$$

A basis for the reciprocal lattice corresponding to the choice Eq. (140) is

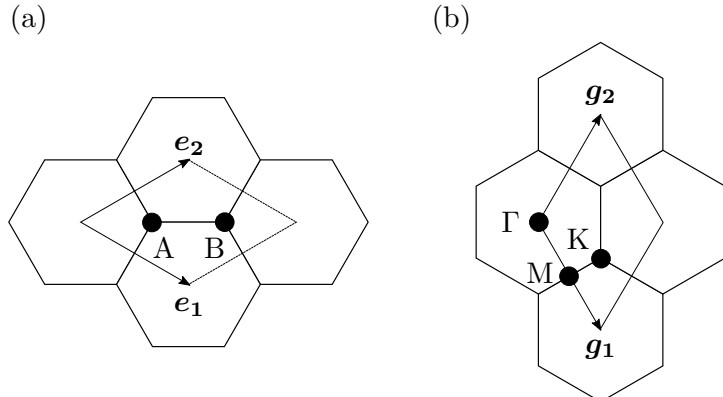

Figure 13: (a) Choice of unit cell for the honeycomb lattice. (b) Reciprocal lattice and Brillouin Zone corresponding to that choice.

$$\begin{aligned} \boldsymbol{g}_1 &= 2\pi(1/\sqrt{3},-1),\\ \boldsymbol{g}_2 &= 2\pi(1/\sqrt{3},1). \end{aligned} \tag{142}$$

The Bravais lattice and reciprocal lattice are shown in Fig. 13. We would like to write a tight-binding Hamiltonian consistent with the symmetries of $p6/mmm$. Defining our tight-binding basis orbitals as

$$\varphi_{\alpha,\boldsymbol{R},s}(\boldsymbol{r}) = \varphi(\boldsymbol{r}-\boldsymbol{R}-\boldsymbol{q}_\alpha)|s\rangle\,, \tag{143}$$

where $\alpha \in \{A,B\}$ denotes the type of the site (often called sublattice), and $|s\rangle$ denotes the spin state $s=\uparrow,\downarrow$ we have the following spin-independent nearest-neighbor hopping:

$$H = t\sum_{\boldsymbol{R},s}\left[c^\dagger_{Bs\boldsymbol{R}}c_{As\boldsymbol{R}} + c^\dagger_{Bs\boldsymbol{R}-\boldsymbol{e}_2}c_{As\boldsymbol{R}} + c^\dagger_{Bs\boldsymbol{R}-\boldsymbol{e}_1}c_{As\boldsymbol{R}}\right] + h.c. \tag{144}$$

Fourier transforming the creation and annihilation operators through the relation

$$c_{ks\alpha} = \sum_{\boldsymbol{R}} e^{-i\boldsymbol{k}\cdot(\boldsymbol{R}+\boldsymbol{q}_\alpha)}c_{\alpha s\boldsymbol{R}} \tag{145}$$

yields the following matrix expression for the hopping term:

$$H(\boldsymbol{k}) = \begin{pmatrix} 0 & Q(\boldsymbol{k}) \\ Q^\dagger(\boldsymbol{k}) & 0 \end{pmatrix} \otimes s_0\,, \tag{146}$$

where $Q(\boldsymbol{k}) = t\left[e^{-i(k_1+k_2)/3} + e^{i(2k_1-k_2)/3} + e^{i(2k_2-k_1)/3}\right]$ (here, $k_1$ and $k_2$ are components of $\boldsymbol{k}$ along the directions of $\boldsymbol{g}_1$ and $\boldsymbol{g}_2$, correspondingly), and $s_0$ is the $2\times 2$ identity matrix in the space of spins. Let us focus in particular at the high-symmetry points $\Gamma = (0,0)$, $M = 2\pi(1/2,0)$ and $K = 2\pi(2/3,1/3)$ in the Brillouin zone (given in reduced coordinates). At these points, the Hamiltonian reduces to

$$\begin{aligned} H(\Gamma) &= 3t\sigma_x \otimes s_0\,,\\ H(M) &= (t/2\sigma_x + \sqrt{3}/2\,t\sigma_y)\otimes s_0\,,\\ H(K) &= 0\,. \end{aligned} \tag{147}$$

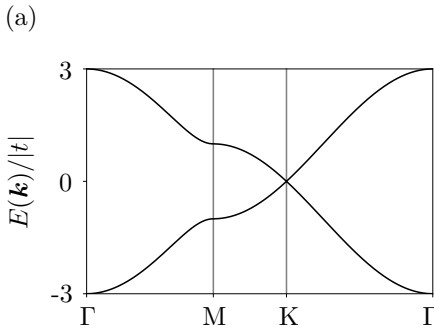
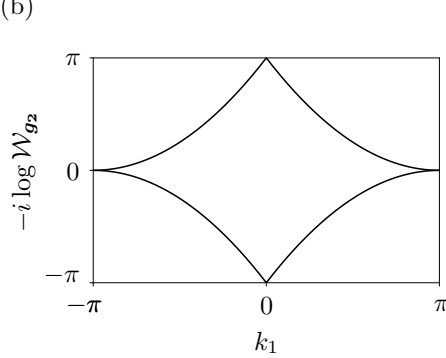

Figure 14: (a) Bands corresponding to the Hamitonian in Eq. (144), with the characteristic Dirac cone at K. (b) Eigenvalues of $W_{g_2}(k_1)$, after opening a gap at K with the term $H_{SO}$ from Eq. (152)

We have introduced the Pauli matrices $\vec{\sigma}$ which act in the basis of $A, B$ sublattice states. Thus, $H(\Gamma)$ and $H(M)$ are gapped, while $H(K)$ (and its time-reversed partner $H(K')$) has a linearly dispersing fourfold degenerate Dirac point at $E = 0$. These features can be seen in Fig. 14a.

Inversion symmetry is represented as $\sigma_x \propto h(\Gamma)$ at the $\Gamma$ point. Let us illustrate the derivation of the matrix for inversion at $M = (1/2, 0)$. We denote by $|\chi_A(\boldsymbol{k})\rangle, |\chi_B(\boldsymbol{k})\rangle$ the Fourier-transformed basis orbital states at $A$ and $B$ lattice sites, respectively. Using the fact that the matrix representation of inversion in the $|\chi_A\rangle, |\chi_B\rangle$ space is $\sigma_x$ we have for the nonzero matrix elements

$$
\begin{aligned}
I_{AB}(M) &= \langle \chi_A(-M)|\hat{I}|\chi_B(M)\rangle = \langle \chi_A(-M)|\chi_A(M)\rangle \\
&= e^{i\boldsymbol{g}_1 \cdot \boldsymbol{q}_A} \langle \chi_A(M)|\chi_A(M)\rangle = e^{i\boldsymbol{g}_1 \cdot \boldsymbol{q}_A},
\end{aligned}
\tag{148}
$$

and

$$
\begin{aligned}
I_{BA}(M) &= \langle \chi_B(-M)|\hat{I}|\chi_A(M)\rangle = \langle \chi_B(-M)|\chi_B(M)\rangle \\
&= e^{i\boldsymbol{g}_1 \cdot \boldsymbol{q}_B} \langle \chi_B(M)|\chi_B(M)\rangle = e^{i\boldsymbol{g}_1 \cdot \boldsymbol{q}_B}.
\end{aligned}
\tag{149}
$$

Thus, at $M$ the matrix for inversion is:

$$
I(M) = \begin{pmatrix} 0 & e^{i2\pi/3} \\ e^{i4\pi/3} & 0 \end{pmatrix} \otimes s_0 = -(1/2\sigma_x + \sqrt{3}/2\sigma_y) \otimes s_0,
\tag{150}
$$

which is proportional to and commutes with $H(M)$. After simultaneously diagonalizing $H(M)$ and $I(M)$, we conclude that, while the lowest bands at $\Gamma$ have inversion eigenvalues $(-, -)$, at $M$ they have inversion eigenvalues $(+, +)$. Based on these inversion eigenvalues and TR symmetry, we can determine[13] the eigenvalues of the Wilson loop matrix $\mathcal{W}_{g_2}(k_1)$ at $k_1 = 0$ and $k_1 = \pi$:

$$
\begin{aligned}
W_{g_2}(0) &= -\sigma_0, \\
W_{g_2}(\pi) &= \sigma_0,
\end{aligned}
\tag{151}
$$

where the degeneracy is due to $T^2 = -1$ (or in this case, simply spin conservation).

---

[13]The proof falls out of the range of these notes, but it can be found in Refs. [21, 26]

If we could gap the Dirac points at $K, K'$ while preserving TR symmetry, in such a way that the lower two bands form an isolated set, the Wilson loop spectrum of this set would be the one shown in Fig. 14b. If we divide the hybrid Wannier functions in a TR symmetric way, each hybrid Wannier function center would wind, leading to a Wannier obstruction. However, in this case we could sacrifice TR symmetry to form non-winding hybrid Wannier functions that do not transform locally under TR symmetry. Hence this phase is protected by TR symmetry. Also, since crossings in the spectrum of $W_{g_2}(k_1)$ are protected only at $\Gamma$ and $M$, the Wilson loop can generically either wind once or not at all, meaning that we can characterize the phases by a $\mathbb{Z}_2$ **invariant**.

We need to show that we can open such a gap at $K, K'$, without breaking TR symmetry. As Kane and Mele showed [27], this requires spin-orbit coupling, which can be included via the following term:

$$H_{so} = -i\lambda \sum_{\ll RR' \gg} s_z^{\sigma\sigma'} \nu_{RR'} \left( c_{AR\sigma}^\dagger c_{AR'\sigma'} + c_{BR\sigma}^\dagger c_{BR'\sigma'} \right). \tag{152}$$

Here $s_z$ is the $z-$directed Pauli matrix in the basis of spin states, $\nu_{RR'} = (d_1 \times d_2)_z / |d_1 \times d_2|$, where $d_1$ and $d_2$ are the nearest-neighbor vectors along the bonds that the electron should traverse to go from the site in $R'$ to the site in $R$. In particular, at the $K$ points, we have:

$$H(K) = \begin{pmatrix} s_z & 0 \\ 0 & -s_z \end{pmatrix}, \tag{153}$$

thus, the desired gap is opened by adding the spin-orbit term.

Let us conclude with a note about the role of inversion symmetry. Even though inversion symmetry allowed us to deduce the $\mathbb{Z}_2$ invariant characterizing this phase, it is not necessary for protecting the topology: the Wannier obstruction needs only TR symmetry. Without inversion symmetry, however, we need to do more work to deduce that the Kane-Mele model is topologicaly nontrivial.

# 6 Exercises

1. Consider a parametric family of Hamiltonians $H(\lambda)$ with discrete spectrum. Show that the projector $P(\lambda)$ onto the $N$ states with energies $\{E_n(\lambda) | n = 0, 1 \dots, N\}$ can be written as

$$2\pi i P(\lambda) = \oint_{\mathcal{C}} dz [z - H(\lambda)]^{-1}, \tag{154}$$

   where $z$ is a complex variable, and $\mathcal{C}$ is a contour enclosing all the $E_n(\lambda)$, and no other eigenvalues of $H(\lambda)$.

2. Given a Hermitian projector $P(t)$ that depends on some parameter $t$, show that

$$P\dot{P}P = 0. \tag{155}$$

3. Show that the adiabatic evolution operator $U_A(t)$ satisfies Kato's equation

$$\dot{U}_A = [\dot{P}, P]U_A. \tag{156}$$

4. Given a basis $\{|\psi_m(\lambda)\rangle\}$ of Im($P$), show that the matrix elements of the adiabatic evolution operator $U_A(t)$ can be written as

$$\langle \psi_n(\lambda) | U_A(\lambda(t)) | \psi_m(0) \rangle = W_{nm}(\lambda), \tag{157}$$

where $W_{nm}(\boldsymbol{\lambda})$ is the path-ordered exponential of the Berry connection along the path $\boldsymbol{\lambda}(t)$.

5. Prove directly that

$$P(\boldsymbol{\lambda})U_A[\boldsymbol{\lambda}(t)]P(0) = \lim_{\delta\boldsymbol{\lambda}\to 0} P(\boldsymbol{\lambda})P(\boldsymbol{\lambda}-\delta\boldsymbol{\lambda})\dots P(\delta\boldsymbol{\lambda})P(0). \tag{158}$$

6. Let

$$\mathfrak{P} = e^{2\pi i x/L}. \tag{159}$$

Show that

$$\mathfrak{P}c_{\boldsymbol{r}}\mathfrak{P}^{-1} = e^{-2\pi i \boldsymbol{r}/L}c_{\boldsymbol{r}}. \tag{160}$$

7. Prove that

$$\sum_{n=1}^{N}\sum_{\boldsymbol{k}}|\psi_{n\boldsymbol{k}}\rangle\langle\psi_{n\boldsymbol{k}}| = \sum_{n=1}^{N}\sum_{\boldsymbol{R}}|w_{n\boldsymbol{R}}\rangle\langle w_{n\boldsymbol{R}}|, \tag{161}$$

where $|\psi_{n\boldsymbol{k}}\rangle$ are the Bloch eigenstates, and $|w_{n\boldsymbol{R}}\rangle$ are the corresponding Wannier functions.

8. Let $\mathcal{C} = \{\gamma(t), t \in [0,1]\}$ be a curve in parameter space (such as the Brillouin zone). Prove that the holonomy $\mathcal{W}_{\mathcal{C}}$ evaluated along the curve satisfies

$$\mathcal{W}_{\mathcal{C}}^{\dagger} = \mathcal{W}_{\mathcal{C}^{-1}}, \tag{162}$$

where the curve $\mathcal{C}^{-1}$ is defined by the function $\gamma(1-t)$. Hence prove directly that the holonomy is a unitary matrix when restricted to $\text{Im}(P)$. Hint: Recall that $PdPP = 0$.

9. Prove that with inversion symmetry, that when $\boldsymbol{k}_\perp \equiv -\boldsymbol{k}_\perp$ that the eigenvalues of $\mathcal{W}_{\boldsymbol{g}}(\boldsymbol{k}_\perp)$ are either real or come in complex conjugate pairs.

10. We denote by "tight-binding limit" the situation in which the position operator is diagonal in the basis of orbitals:

$$\langle\varphi_{\alpha\boldsymbol{R}}|r|\varphi_{\beta\boldsymbol{R}'}\rangle = (\boldsymbol{R}+\boldsymbol{t}_\alpha)\delta_{\alpha\beta}\delta_{\boldsymbol{R}\boldsymbol{R}'}, \tag{163}$$

where $\varphi_{\alpha\boldsymbol{R}}(\boldsymbol{r})$ and $\varphi_{\beta\boldsymbol{R}'}(\boldsymbol{r})$ are orthonormal orbitals centered at positions $\boldsymbol{R}+\boldsymbol{t}_\alpha$ and $\boldsymbol{R}'+\boldsymbol{t}_\beta$, respectively. $\boldsymbol{R}$ and $\boldsymbol{R}'$ denote lattice vectors, while $\boldsymbol{t}_\alpha$ and $\boldsymbol{t}_\beta$ are vectors within the unit cell.

Show that in the tight-binding limit the Berry connection can be calculated from the coefficients of the expansion of eigenstates of the Hamiltonian in terms of Bloch functions constructed from the basis orbitals. In particular, show that in the tight-binding limit in one dimension:

$$\log\langle\mathfrak{P}\rangle = -\text{Tr}\left[\oint dk\langle u_{nk}|\partial_k u_{mk}\rangle\right]. \tag{164}$$

11. Recall that the Bloch Hamiltonian for the Rice-Mele chain in terms of the $\sigma = s, p$ basis functions $|\sigma\boldsymbol{R}\rangle$ is

$$h(k) = (\epsilon + 2t\cos k)\sigma_z + 2t\sin k\sigma_y. \tag{165}$$

Compute the Wannier functions for this model when a) $t = 0$ and b) $\epsilon = 0$.

12. Let $s = \{\mathcal{R}|0\}$ be a symmetry such that

$$\begin{aligned}\mathcal{R}\boldsymbol{k}_\perp &\equiv \boldsymbol{k}_\perp, \\ \mathcal{R}\boldsymbol{g} &= \boldsymbol{g}.\end{aligned} \tag{166}$$

Show that in this case the eigenstates of the Wilson loop $\mathcal{W}_{\boldsymbol{g}}(\boldsymbol{k}_\perp)$ can be labelled by their eigenvalues under the operator $U_{\mathcal{R}}$.

13. Show that for a time-reversal symmetric system that:

    (a) $\mathcal{W}_g(\boldsymbol{k}_\perp)$ and $\mathcal{W}_g(-\boldsymbol{k}_\perp)$ are isospectral.
    (b) If $\mathcal{T}^2 = -1$ then eigenstates of $\mathcal{W}_g(\boldsymbol{k}^*)$ at TRIMs $\boldsymbol{k}^*$ are doubly degenerate.

14. Prove the Ambrose-Singer theorem in the case of a single band (i.e. $\operatorname{rank} P = 1$)

15. Prove for a 1D system that $\det W_g = \pm 1$ is determined by the parity of the occupied states at $\Gamma$ and $X$. Hint: Use the fact that $I \mathcal{W}_{0\leftarrow -\pi} I^{-1} = \mathcal{W}_{0\leftarrow \pi}$

## 7 Conclusion

After making it this far, we hope the reader has come away with a renewed appreciation for the role of geometric transport in condensed matter physics. With our unorthodox organization of the material, we sought to highlight some oft-overlooked connections between geometry and topology. As mentioned above, a more comprehensive treatment of these topics can be found in Refs. [2, 10]. Furthermore, for those interested in exploring more applications of these methods to topological insulators, we recommend Refs. [4–6, 8, 28, 29] as a starting point.

## Acknowledgements

Mikel Iraola acknowledges support by the Spanish Ministerio de Ciencia e Innovacion (grant number PID2019-109905GB-C21). Barry Bradlyn acknowledges the support of the Alfred P. Sloan foundation, and the National Science Foundation under grant DMR-1945058.

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
