# Peer review of "Lecture Notes on Berry Phases and Topology"

_SciPost Physics Lecture Notes, doi:SciPost Phys. Lect. Notes 51 (2022)_

## Round 1 · Referee Report · Anonymous (Referee 1) · 2022-1-12

Strengths

1- Mathematically rigorous and original introduction into geometrical and topological aspects of the electronic band theory; 2- Concise exposition; 3- A list of 15 problems that will help an interested and diligent reader to acquire practical knowledge of the material covered in these lectures.

Weaknesses

1- The advanced nature of the presentation makes these notes inaccessible to a broader community of scientists interested in topological aspects of the band theory.

Report

I believe that these lecture notes by Bradlyn and Iraola, targeted at mathematically-inclined readers with a taste for rigorous arguments, will serve as an excellent introduction into the field of topological electronic band structures. The notes present the key notions of the Berry phase, polarization, Wilson loops, Wanier functions and the Chern number using rigorous, well-thought-through arguments. Moreover, the authors illustrate these main concepts with a help of several paradigmatic models. For an ambitious and invested reader, in the end of the main text, the authors collected 15 problems to be solved. I believe that the lecture notes are timely. I recommend publication in SciPost Physics Lecture Notes in the present form.

Requested changes

I have only minor comments: 1- While most figures are mentioned in the text, I believe that there is no reference to Fig. 1 in the main text. I suggest adding a reference to Fig.1 in the main text. 2- What is $\Delta \tau$ in the second paragraph of page 2 and why should it be much larger than unity? 3- In the caption of Fig. 3 the notation in the last sentence is confusing. Probably $\dots(\pi/2, 2\pi t)/t\in\dots$ should be replaced with $\dots(\pi/2, 2\pi t)|t\in\dots$. 4- There is a typo "spectrum spectrum" in the first sentence of Section 5.

---

## Round 1 · Referee Report · Anonymous (Referee 2) · 2022-2-1

Strengths

  1. The notes are very clear and well-written.

  2. The manuscript is very accessible. It captures all the nuances of different topological phases without introducing any unnecessary complications.

  3. Presenting the topic from the point of view of Wannier function and hybrid Wannier functions was a good choice.

  4. The examples presented in the notes are enlightening, specially the discussion of the Rice-Mele model (subsection 4.1).

  5. The discussion of discrete symmetries is very clear and do not rely on crystallography jargons.

Weaknesses

  1. The discussion about Berry Phase and Polarization (section 3) could be improved.

  2. The quantization of the Hall conductivity was a bit overlooked.

Report

In these lecture notes, the authors address Berry phase effects in the electronic structure of solids. One of the key points in these notes is to introduce the geometrical structure behind the adiabatic theorem and apply these ideas to the band theory of insulators, which is the framework behind the theory of topological band insulators. I am fairly pleased with the presentation as well as the content of the manuscript, and I believe that these notes should certainly be published in Scipost. Nevertheless, I believe that some parts of the notes can be made a little more accessible for the readers who had no contact with the subject before. For simplicity, I will break down my comments and suggestions per section.

Section II:

Here, the authors present the adiabatic theorem from a more geometrical viewpoint. They managed to present the main ideas without digressing too much into the theory of fiber bundles, which makes the presentation very clear. In the following, I will list some of my comments and suggestions.

  1. The introduction of the projection operator  might be a bit abstract for first time readers. It is perhaps worthwhile to motivate the introduction of  $P(\boldsymbol \lambda)$  from the Solid State perspective. Anticipating that they represent the filled energy bands in an insulator, either in a sentence or a footnote.

  2. In p. 05, I suggest writing $\tau\Delta\gg 1$  instead of $\Delta\tau\gg1$. It avoids any possible confusion, such as, interpreting $\Delta\tau$ as the increment of $\tau$.

  3. In the sentence "Since we are interested primarily in the behavior of the subspace  , however, ..." (p. 05), I would replace the word "however" by "here" or "in this section".

  4. Eq. (15) is parametrization invariant since it depends only on the endpoints of the corresponding curve in the parameter space . However, the main assumption of adiabatic theorem is that the system must evolve slowly. Obviously, as soon as we choose $f(H(\boldsymbol \lambda))=H(\boldsymbol \lambda)$, the condition $\tau\Delta\gg1$  fixes the parametrization. I would like to see a small discussion (few words) on why slow evolution is still necessary even though the geometric phase is parametrization invariant.

Section III:

In this section, the authors address the polarization of an electronic chain and connect it to the Berry phase. Sadly, this section did not keep the high quality of the other ones. It is perhaps too short and very much calculation driven. Below I list some points that could improve it.

  1. It is common knowledge, but I still believe that the authors should indicate that the crystal momenta is defined up to $2\pi/a$, that is,  $k\sim k+2\pi/a$, and that the manifold $\mathcal M$  is isomorphic to the unit circle $S^1$.  These points are somehow glimpsed in footnote 4, however, I think they should be spelled out in more detail throughout the text.

  2. Whoever had a crash course in Solid State Physics knows that  $|qE_0|\ll\Delta$ means the electric field is not strong enough to introduce transitions between the valence and the conduction band. However, I believe it is worthwhile to repeat such information in this section. This provides a link between old concepts in solid state and adiabatic evolution.

  3. It is not clear in the definition of  $A_{mn}$ (at the bottom of p. 12) that the integration domain is the unit cell. 

  4. The last equality of Eq. (62) is not very clear for beginners. Since these are lecture notes, I think the authors could explain how to connect Eq. (62) with the expression (30) in a slightly more detailed way. Also, both the time $t$  and the crystal momentum  $k$ are dimensional parameters. Writing $W(2\pi)$  in Eq. (62) is somewhat confusing. 

  5. It must be $k+2\pi/a$ instead of $k+2\pi$ in footnote 4. So far, the authors did not indicate they were setting $a=1$.

  6. I personally think the simplest way to explain that the electronic polarization is only defined up to a multiple of $ea$, derives from the fact that the polarization itself is only well-defined for electric neutral systems. Considering the origin $\boldsymbol 0$ and $\boldsymbol R$ to be inside the domain $V$, the polarization measured from the origin becomes ${\bf P}{{\bf 0}}=\int_V\rho ({\bf r}-{\bf R}+{\bf R}) \,d^Dr=\bf P)$}+{\bf R} \times (\text{total charge in V. From this, we see that even though the sample polarization is well-defined (electric neutral), the electronic polarization is not. Setting $D=1$ and $\boldsymbol R$ to be a point in the 1D Bravais lattice, we find that the electronic polarization becomes $P_0=P_{na}-na(N_e e)$, where $N_e$ is the number of electrons in $V$. Moreover, we see that $(nN_e) ea$ is nothing but the polarization of the ionic chain. Some discussion in these lines might be useful to the reader. 

Section IV:

In this section, the authors introduce the Wannier functions and study their properties through an example (Rice-Mele model). I am very pleased with this section, in particular with the discussion in subsection 4.1. The comparison between the flat band cases is very clear and well-presented. It captures all the nuances of different topological phases without introducing any unnecessary complications. In the following are minor comments about the section, since I believe this is good as it is.

  1. The index under the summation sign in the Eq. (72) should be $m$ instead of $n$.

  2. The authors could have cited an appropriate reference after the equation $2\pi i n=\oint Tr[U^\dagger(k)\partial_k U(k)] dk$ at p. 16 for further reading about winding numbers in topological systems.

  3. In the first integral of Eq. (92), the domain should be the unit cell.

Section IV:

In this section, the authors address Wilson loops and their connection to the Chern number and the non-analyticity of the Bloch functions. In addition to that, they study how symmetries constrain the operator spectrum. At the end, they apply the discussion in two cases, namely, the Thouless Pump and the Kane-Mele model. This section is indeed more dense than the previous ones. Yet, the exposition here is quite clear. I particularly enjoyed the discussion in subsection 5.2. 

Similarly as in the section 4, here I will just list some typos I could find in the text.

  1. The word spectrum is duplicated in the first sentence of this section.

  2. The expression below Eq. (106) should be $|u'{n\boldsymbol k}\rangle=U^\dagger\rangle$}(\boldsymbol k)|u_{m\boldsymbol k instead of $|u'{n\boldsymbol k}\rangle=U^\dagger\rangle$}(\boldsymbol k)|u_{n\boldsymbol k.

In summary, these are optional considerations and the authors are free to accept the ones they find relevant. The only point I feel strongly about is that the authors improve section 3, even slightly, so it keeps up with the rest of the manuscript.

---

## Editorial Decision

published